



# Evaluating permafrost physics in the CMIP6 models and their sensitivity to climate change

Eleanor Burke[1], Yu Zhang[2], and Gerhard Krinner[3]

[1]Met Office Hadley Centre, FitzRoy Road, Exeter, EX1 3PB, UK.
[2]Natural Resources Canada: Ottawa, Ontario, Canada.
[3]Institut des Géosciences de l'Environnement, CNRS, Université Grenoble Alpes, Grenoble, France.

**Correspondence:** Eleanor Burke (eleanor.burke@metoffice.gov.uk)

**Abstract.** Permafrost is an important component of the Arctic system and its future fate is likely to control changes in northern high latitude hydrology and biogeochemistry. Here we evaluate the permafrost dynamics in the global models participating in the Coupled Model Intercomparison Project (present generation - CMIP6; previous generation - CMIP5) along with the the sensitivity of permafrost to climate change. Whilst the northern high latitude air temperatures are relatively well simulated by

the climate models, they do introduce a bias into any subsequent model estimate of permafrost. Therefore evaluation metrics are defined in relation to the air temperature. This paper shows the climate, snow and permafrost physics of the CMIP6 multi-model ensemble is very similar to that of the CMIP5 multi-model ensemble. The main difference is that a small number of models have demonstrably better snow insulation in CMIP6 than in CMIP5 which improves their representation of the permafrost extent. The simulation of maximum summer thaw depth does not improve between CMIP5 and CMIP6. We suggest that

models should include a better resolved and deeper soil profile as a first step towards addressing this. We use the annual mean thawed volume of the top 2m of the soil defined from the model soil profiles for the permafrost region to quantify changes in permafrost dynamics. The CMIP6 models suggest this is projected to increase by 20-30 % / $^{o}$C of global mean temperature increase. Under climate change and in equilibrium this may result in an additional 80-120 Gt C / $^{o}$C of permafrost carbon becoming vulnerable to decomposition.

# 1  Introduction

Permafrost, defined as ground that remains at or below 0°C for two or more consecutive years, underlies 22 % of the land in the Northern Hemisphere (Obu et al., 2019). Permafrost temperatures increased by 0.29°C ± 0.12°C between 2007 and 2016 when averaged across polar and high-mountain regions (Pörtner et al., 2019; Biskaborn et al., 2019). This unprecedented change will have consequences for northern hydrological and biogeochemical cycles. For example, it will impact the global

climate (Burke et al., 2017b), ecosystems (Vonk et al., 2015) and fire (Wotton et al., 2017) as well as man made infrastructures (Melvin et al., 2017; Hjort et al., 2018) and human health (D'Costa et al., 2011); leading to issues with the overall sustainability of northern communities (Larsen et al., 2014). The latest generation of the Coupled Model Intercomparison Project [CMIP6 - Eyring et al. (2016)] provides an opportunity to increase our understanding of these potential impacts under future climate change.



CMIP6 provides a coordinated set of Earth System Model simulations designed, in part, to understand how the earth system responds to forcing and to make projections for the future. Here we derive and apply a set of metrics to benchmark the ability of the coupled CMIP6 models to represent permafrost physical processes. Biases in the simulated permafrost arise from (1) biases in the simulated surface climate and (2) biases in the underlying land surface model. Where possible this paper

isolates the land surface component from the surface climate, and focuses on the land surface component. Both Koven et al. (2013) and Slater et al. (2017) evaluated the previous generation of global climate models (CMIP5) and found that the spread of simulated present-day permafrost area within that ensemble is large and mainly caused by structural weaknesses in snow physics and soil hydrology within the some of models. Here we assess any improvements in the CMIP6 multi-model ensemble over the CMIP5 multi-model ensemble. Koven et al. (2013) and Slater and Lawrence (2013) also found a wide variety of

permafrost states projected by the CMIP5 multi-model ensemble in 2100. We question whether the sensitivity of permafrost to climate change is different in this current generation of CMIP models.

Permafrost dynamics can be described by the mean annual ground temperature ($MAGT$) and the maximum thickness of the near surface seasonally thawed layer (the active layer or $ALT$). To first order and at large scale the presence of permafrost is controlled by the mean annual air temperature ($MAAT$). In general if the $MAAT$ is less than $0°C$ there is a chance of

finding permafrost. This is modulated by the seasonal cycle of air temperature (Karjalainen et al., 2019), snow cover, topography, hydrology, soil properties and vegetation (Chadburn et al., 2017). In winter the snow cover insulates the soil from cold air temperatures causing the soil to be being warmer than the air (winter offset, Smith and Riseborough (2002)). In summer any vegetation present should insulate the soil from warm air temperatures and cause the air to be warmer than the soil

(summer offset, Smith and Riseborough (2002)). The thermal offset between the soil surface and the top of the permafrost is mainly due to the seasonal changes of the thermal conductivity between the soil surface and the top of the permafrost - the top of the permafrost tends to be slightly colder than the soil surface temperature (Smith and Riseborough, 2002). Figure 1 shows a schematic of this climate-permafrost relationship which was parameterised by (Smith and Riseborough, 2002; Obu et al., 2019). In fact Obu et al. (2019) developed a large-scale and high resolution observations-based estimate of mean annual

ground temperature and probability of permafrost using this framework.





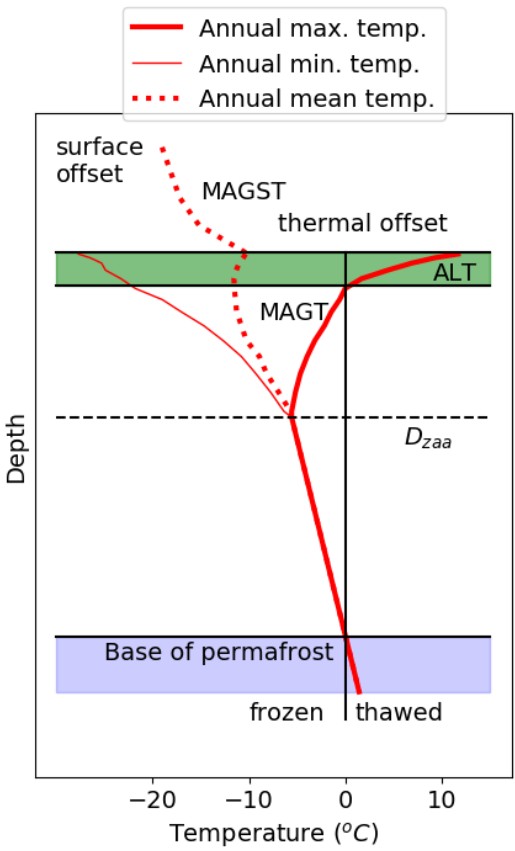

**Figure 1.** Schematic of the mean, minimum and maximum annual temperature profile from the surface boundary layer to below the bottom of the permafrost. $MAAT$ is the mean annual air temperature, $MAGST$ is the mean annual ground surface temperature; $MAGT$ is the temperature at the top of the permafrost; $ALT$ is the maximum depth the permafrost thaws to in any given year. The surface offset is the difference between the $MAAT$ and the $MAGST$ and the thermal offset is the difference between the $MAGST$ and the $MAGT$.

The summer thaw depth depends strongly on the incoming solar radiation as well as soil moisture, soil organic content and topography and responds to short term climate variations (Karjalainen et al., 2019). In particular, soils with a higher ice content thaw more slowly than those with lower ice content resulting in a shallower maximum thaw depth or $ALT$. Under increased global mean temperature gradual thaw will occur increasing both $ALT$ and the time over which the near surface soil is thawed. These two factors can be represented jointly by the annual mean thawed fraction of the soil (Harp et al., 2016), which can also be used as a proxy for the soil carbon exposure to decomposition. Abrupt thaw processes will also occur where landscape dynamics change in a hard-to-predict manner (Turetsky et al., 2019) but are not currently represented in Earth System Models and are not assessed here.





This paper evaluates the ability of the CMIP6 models to represent present-day permafrost dynamics in terms of the presence or absence of permafrost, the mean annual ground temperature ($MAGT$), the maximum active layer thickness ($ALT$) and the annual mean thawed fraction. This is accompanied by an analysis of the improvement in the models' structure when compared with the CMIP5 multi-model ensemble. The assessed soil diagnostics can be used to estimate the large-scale permafrost extent and the annual mean frozen/thawed volume of soil. Finally the simulated sensitivity to climate change of northern high latitude

soils is quantified in order to explore their potential fate in the future and any consequent climate impacts.

## 2  Materials and Methods

### 2.1  CMIP model data

Historical and future monthly mean data were retrieved for a subset of coupled climate models from the CMIP6 (Eyring et al. (2016), Table 1) and the CMIP5 (Taylor et al. (2012), Table S2.1) model archive. The historical simulations run from

1850/1860 through to the end of 2014 (CMIP5 - to end of 2005). The CMIP5 future simulations are based on Representative Concentration Pathways (RCPs, Taylor et al. (2012)) which combine scenarios of land use and emissions to give a range of future outcomes through to 2100. When available RCP8.5 (high pathway), RCP4.5 (intermediate pathway) and RCP2.6 (peak and decline pathway) are used here. The future simulations used from CMIP6 (O'Neill et al., 2016) combine socioeconomic trends with RCP scenarios and are based on the following Shared Socioeconomic Pathways (SSPs): SSP5-8.5 (high end of

the range of future pathways and updating RCP8.5), SSP3-7.0 (medium to high end of the range and an unmitigated forcing scenario), SSP2-4.5 (medium part of the range and updating RCP4.5), and SSP1-2.6 (low end and updating RCP2.6).

Monthly diagnostics processed are surface air temperature (tas; equivalent to 2-m temperature), snow depth (snd), and vertically resolved soil temperatures (tsl) for latitudes greater than 20°N for the first ensemble member of each model (i.e.,

simulation r1i1p1 or similar). Each model is left at its native grid. In addition, grid cells with exposed ice or glaciers at the start of the historical simulation are masked out and the land fractions in the models are accounted for in any area-based assessment of permafrost.



| Model | Institiute | Land model | No. layers | Soil depth (m) | $D_{zaa}$ (m) |
|---|---|---|---|---|---|
| BCC-CSM2-MR | BCC | BCC_AVIM2 | 10 | 2.9 | - |
| BCC-ESM1 | BCC | BCC_AVIM2 | 10 | 2.9 | - |
| CESM2 | NCAR | CLM5 | 25 | 42.0 | 19.4 |
| CESM2-WACCM | REF | CLM5 | 25 | 42.0 | 19.9 |
| CNRM-CM6-1 | CNRM-CERFACS | Surfex 8.0c | 14 | 10.0 | - |
| CNRM-ESM2-1 | CNRM-CERFACS | Surfex 8.0c | 14 | 10.0 | - |
| CanESM5 | CCCma | CLASS3.6/CTEM1.2 | 3 | 4.1 | - |
| EC-Earth3 | EC-Earth-Consortium | HTESSEL | 4 | 1.9 | - |
| GFDL-CM4 | NOAA-GFDL | GFDL-LM4.0.1 | 20 | 8.8 | - |
| GISS-E2-1-G | NASA-GISS | GISS LSM 6 | 2.7 | - | |
| GISS-E2-1-H | NASA-GISS | GISS LSM 6 | 2.7 | - | |
| HadGEM3-GC31-LL | MOHC, NERC | JULES-HadGEM3-GL7.1 | 4 | 2.0 | - |
| IPSL-CM6A-LR | IPSL | ORCHIDEE (v2.0, Water /Carbon/Energy mode) | 18 | 65.6 | 16.0 |
| MIROC6 | MIROC | MATSIRO6.0 | 6 | 9.0 | - |
| MIROC-ES2L | MIROC | MATSIRO6.0+ VISIT-e ver.1.0 | 6 | 9.0 | - |
| MPI-ESM1-2-HR | MPI-M, DWD DKRZ | JSBACH3.20 | 5 | 7.0 | - |
| MRI-ESM2-0 | MRI | HAL 1.0 14 | 8.5 | - | |
| NorESM2-LM | NCC | CLM | 25 | 42.0 | 18.9 |
| UKESM1-0-LL | MOHC, NERC, NIMS-KMA, NIWA | JULES-ES-1.0 | 4 | 2.0 | - |

**Table 1.** A summary of the CMIP6 models used in this study including the number of soil layers and the depth of the middle of the bottom soil layer. Also show is $D_{zaa}$ for the models where the difference in the annual maximum and minimum soil temperatures at the maximum soil depth is less than $0.1^\circ$C. CMIP5 models are summarised in Table S2.1.

## 2.2 Observational-based data sets

**2.2.1 Air temperature**

Air temperature observations over land at 2 m were taken from the WATCH Forcing Data methodology applied to ERA-Interim (WFDEI) data set (Weedon et al., 2014). Monthly corrections generated from Climate Research Unit (CRU) (Mitchell and Jones, 2005) were applied to the Era-Interim reanalysis data (ECMWF, 2009). Air temperatures are available at $0.5^\circ$ reso-



lution and were aggregated to monthly and annual means.


### 2.2.2 Large scale snow depth product

This data set consists of a Northern Hemisphere subset of the Canadian Meteorological Centre (CMC) operational global daily snow depth analysis Brown and Brasnett (2010). The analysis is performed using real-time, in-situ daily snow depth observations, and optimal interpolation with a first-guess field generated from a simple snow accumulation and melt model driven with

temperatures and precipitation from the Canadian forecast model. The analysed snow depths are available at approximately 24 km resolution for the period between 1998 and 2016 and were converted from daily to monthly means.

### 2.2.3 Permafrost extent

The International Permafrost Association (IPA) map of permafrost presence (Brown et al., 1998) gives a historical permafrost

distribution compiled for the period between 1960 and 1990. It separates continuous (90-100%), discontinuous (50-90%), sporadic (10-50%), and isolated (<10%) permafrost. This distribution was generated from the original 1:10,000,000 paper map and the version used here was re-gridded to $0.5°$ resolution.

The Climate Change Initiative permafrost (CCI-PF) reanalysis data set (Obu et al., 2019) is a recently developed data set

that supplies the mean annual ground temperature ($MAGT$) and the probability of permafrost for each grid cell. These were derived from an equilibrium model of permafrost at 1 km resolution and provide a snapshot of the 2000–2016 period. The model is driven by remotely-sensed land surface temperatures, down-scaled ERA-Interim climate reanalysis data, tundra wetness classes and landcover map. These data were within $\sim \pm 2°C$ of in situ borehole measurements. This CCI-PF analysis of permafrost extent is within the range but slightly lower than the estimate of Brown et al. (1998). We use a version of the CCI-PF

which has been re-gridded to $0.5°$ resolution.

An alternative way of deriving an observational-based estimate of permafrost presence is to derive the probability of permafrost from the observed mean annual air temperature ($MAAT$). An observational-based relationship was defined by Chadburn et al. (2017) who updated Gruber (2012). The Chadburn et al. (2017) relationship has a 50% chance of the presence

of permafrost at -4.3°C. Using the Chadburn et al. (2017) relationship, we reconstructed a spatial map of the probability of permafrost from the WFDEI estimates of $MAAT$ and estimated model-specific benchmark permafrost distributions ($PF_{benchmark}$). $PF_{benchmark}$ differs from the observations because it is derived from the model $MAAT$ and is a better metric for assessing the land surface module.

Table 2 summarises the different observational-based permafrost distributions. The permafrost affected area ($PF_{aff}$) is defined as the area where the probability of permafrost is greater than 0.01 and is expected to be very similar to the land surface

| Data set | $PF_{aff}$ $(10^6\text{km}^2)$ | $PF_{aff}$/area $MAAT<0°C$ | $PF_{ex}$ $(10^6\text{km}^2)$ | $PF_{ex}$/area $MAAT<-4.3°C$ | $PF_{50\%}$ $(10^6\text{km}^2)$ | $PF_{50\%}$/area $MAAT<0°C$ |
|---|---|---|---|---|---|---|
| Brown et al. (1998) | 24.3 | 0.99 | 17.1 | 1.12 | 18.7 | 0.77 |
| Obu et al. (2019) | 20.0 | 0.82 | 13.6 | 0.89 | 13.5 | 0.55 |
| Chadburn et al. (2017) | 24.7 | 1.01 | 15.3 | 1.00 | 15.1 | 0.62 |

**Table 2.** Permafrost areas from three of the available observational data sets defined both as an areal extent and as a fraction of the area where the observed $MAAT$ is below the given threshold. $PF_{aff}$ is defined as the permafrost affected area and includes any grid cells which have a non-zero probability (>1%) of permafrost occurrence. $PF_{ex}$ is the area of permafrost weighted by the proportion of permafrost in each grid cell and $PF_{50\%}$ is the area where the probability of finding permafrost is $\geqslant 50\%$. The Chadburn et al. (2017) relationship has a 50% chance of the presence of permafrost at -4.3°C. $MAAT$ is the mean for the period 1995 to 2014.

area where the $MAAT$ is less than 0°C. Table 2 shows that this is the case for the Brown et al. (1998) and Chadburn et al. (2017) data set, but the Obu et al. (2019) CCI-PF data set has a slightly lower $PF_{aff}$ area. It should be noted that the Brown et al. (1998) observational data sets was one of the data sets used to develop the Chadburn et al. (2017) relationship but the Obu
et al. (2019) CCI-PF reanalysis data set was developed independently. The permafrost extent ($PF_{ex}$) is the area of permafrost weighted by the proportion of permafrost in each grid cell and $PF_{50\%}$ is the area of the grid cells where the probability of finding permafrost is greater than 0.5. These two definitions produce a very similar land surface area. Overall there is some uncertainty in the proportion of the land surface with $MAAT < 0°C$ that contains permafrost - the observational estimates are 0.55, 0.62 and 0.77. The Brown et al. (1998) data have a similar value for the $PF_{aff}$ area but a higher probability of finding
permafrost at air temperature less than -4.3°C. Overall the CCI-PF data (Obu et al., 2019) shows consistently less permafrost (Table 2).

### 2.2.4 Site specific observations

The Circumpolar Active Layer Monitoring Network [CALM: Brown (1998)] is a network of over 100 sites at which ongoing
measurements of the end of season thaw depth or the $ALT$ are taken. Measurements are available from the early 1990's, when the network was formed, to present.

$MAAT$, $MAGT$, snow depth and $MAGST$ at 20 cm were available for a range of sites in Russia and Canada (Zhang et al., 2018). The data at Russian meteorological stations are from All-Russian Research Institute of Hydrometeorological
Information - World Data Centre (RIHMI-WDC) and data at Canadian climate stations are from Environment and Climate Change Canada. This gives data from ∼330 stations. Additional data is available from the Global Terrestrial Network for Permafrost (GTN-P; Biskaborn et al. (2015, 2019). Ground temperatures are measured in > 1000 boreholes at a wide variety of depths. Years with a complete seasonal cycle were extracted from selected boreholes where the data available includes $MAAT$,





$MAGST$ at 20 cm and $MAGT$. A full description of these data and their post-processing are included in Zhang et al. (2018).


## 2.3 Evaluation metrics

These metrics are derived from both the models and the observations in a consistent manner.

### 2.3.1 Effective snow depth

Snow has a big impact on the soil temperatures and presence/absence of permafrost in the northern high latitudes [Wang et al.

(2016); Zhang (2005)]. Here we use the effective snow depth, $S_{depth,eff}$ (Slater et al., 2017) which describes the insulation of snow over the cold period. $S_{depth,eff}$ is an integral value such that the mean snow depth ($S$ in m) each month ($m$) is weighted by its duration:

$$S_{depth,eff} = \frac{\sum_{m=1}^{M} S_m (M+1-m)}{\sum_{m=1}^{M} m} \qquad (1)$$

It is assumed that the snow can be present anytime from October ($m = 1$) to March ($m = 6$) with the maximum duration, $M$

equal to 6 months. This weights early snowfall more than late snowfall as it will have a greater overall insulating value. The insulation capacity of the snow changes little with snow depth when $S_{depth,eff}$ increases above ~0.25 cm (Slater et al., 2017), and seasons with an earlier snowfall will generally have a greater $S_{depth,eff}$ than seasons with a later snowfall.

### 2.3.2 Winter, summer and thermal offsets

The winter offset is defined as the difference between the mean soil temperature at 0.2 m and the mean air temperature for the

period from December to February. This is expected to be positive with the soil temperature warmer than the air temperature. The summer offset is defined in a similar manner for the period between June and August. This is expected to be slightly negative with the soil temperature cooler than the air temperature. The surface offset is the sum of the summer and winter offset, but is dominated by the winter offset. The thermal offset is the temperature difference between the annual mean soil temperature at 0.2m (mean annual ground surface temperature: $MAGST$) and the annual mean soil temperature at the top

of the permafrost (mean annual ground temperature: $MAGT$). This is expected to be slightly negative with $MAGT$ slightly colder than $MAGST$.

### 2.3.3 Diagnosing permafrost in the model

In this paper the preferred method of defining permafrost is to diagnose the temperature at the depth of zero annual amplitude

($D_{zaa}$). $D_{zaa}$ is defined as the minimum soil depth where the variation in monthly mean temperatures within a year is less than 0.1°C. If the temperature at the $D_{zaa}$ is less than 0°C for a period of 2 years or more there is assumed to be permafrost in that grid cell. However, only 4 of the CMIP6 models have a soil profile deep enough to identify the $D_{zaa}$ (Table 1). In the



remainder of the models, the maximum soil depth is less than the $D_{zaa}$ and an alternative method of identifying the presence of permafrost is required. For these models permafrost is assumed to be present in grid cells where the 2-year mean soil tem-

perature of the deepest model level is less than 0°C. This definition was used by Slater and Lawrence (2013) who suggested that if the mean soil temperature of the deepest model level is below 0°C and assuming constant soil heat capacity, there is likely to be permafrost deeper in the soil profile. However, this definition does not explicitly recognise permafrost in the soil profile - in order to do that, the maximum soil temperature of the deepest model level should be less than 0°C. The main issue with this latter method is that, if the soil profile does not extend deep enough, the deepest model level may fall above $ALT$ and

the permafrost extent will be under-estimated.

Subgrid scale variability is not taken into account in this assessment - the models are assumed either to have permafrost or no permafrost in each grid cell. However, the observations are either very high resolution (CCI-PF is 1 km resolution in its original format) or supply a probability of permafrost for each grid cell (Brown et al., 1998). Therefore in order to compare the ob-

served extent with those from the models, we assume that any grid cell where the observations have $\geqslant 50\%$ permafrost should be identified by the models as having permafrost and any grid cells with $< 50\%$ will be identified as not having permafrost. The observed values of this threshold ($PF_{50\%}$) are shown in Table 2 and are approximately equal to the permafrost extent ($PF_{ex}$, also shown in Table 2) which is defined as the observed area of permafrost which takes into account the proportion of permafrost in each of the grid cells.


### 2.3.4 Thaw depth and associated metrics

The thawed depth from the surface is defined for each month using the depth-resolved monthly mean soil temperatures. The soil temperatures were interpolated between the centre of each model level and the thaw depth defined at the minimum depth where it reaches 0°C. Some of the models have a very poorly resolved soil temperature profile which will introduce some

biases into this estimate (Chadburn et al., 2015). In addition, taliks (unfrozen patches within the frozen part of the soil) will not be identified using this method. The annual maximum active layer thickness ($ALT$ in m) is defined as the maximum monthly thaw depth for that year.

Under increasing temperature both the $ALT$ and the time the soil is thawed will increase via an earlier thaw and later freeze

up. Therefore, Harp et al. (2016) defined the annual mean thawed fraction ($\widetilde{D}$) for permafrost soils which can be expressed in units of m³/m³:

$$\widetilde{D} = \frac{1}{z_{max}} \frac{1}{12} \int\limits_{1}^{12} \int\limits_{0}^{z_{max}} H(T(z,t)) dz dt \qquad H(T(z,t)) = \begin{cases} 1 & \text{if } T(z,t) > 0, \\ 0 & \text{if } T(z,t) <= 0. \end{cases} \qquad (2)$$

where $t$ is the time in months; $z$ is the depth in m; $T$ is the soil temperature at time $t$ and depth $z$; and $z_{max}$ is the maximum depth of the soil under consideration. Here we assume $z_{max}$ is 2 m which is relatively shallow but enables the models with shallower





soil depths to be included consistently within the analysis. The annual mean frozen fraction ($\widetilde{F}$) is the frozen component of the soil and given by:

$$\widetilde{F} = 1 - \widetilde{D} \tag{3}$$

One advantage of using $\widetilde{D}$ over $ALT$ is that it enables taliks to be identified. In addition, $\widetilde{D}$ is a first order proxy for the soil carbon exposure to decomposition in any particular grid cell.

The annual thawed volume ($\widetilde{D}_{tot}$ in $m^3$) is the sum of the area-weighted values of $z_{max}\widetilde{D}$ for each grid cell in the present day permafrost region. Any non-permafrost grid cells are masked. Similarly the annual frozen volume ($\widetilde{F}_{tot}$ in $m^3$) is the sum of the area-weighted $z_{max}\widetilde{F}$ for each grid cell again defined for the present day permafrost region. For any future projections if there is no longer freezing in a specific grid cell (which had permafrost in the present day) $\widetilde{D}$ is set to 1 and $\widetilde{F}$ is set to 0.

We can derive an observational-based estimate of $\widetilde{D}_{tot}$ using the available site specific data described in Section 2.2.4 and the CCI-PF dataset. $\widetilde{D}$ is related non-linearly to the $MAGT$ - the warmer the ground temperature the bigger the annual mean thawed fraction. A second order polynomial was fitted to the site-specific relationship between $\widetilde{D}$ and the $MAGT$ and then used in conjunction with the in the CCI-PF data set to derive $\widetilde{D}$ for each grid cell with permafrost present. Summing over the CCI-PF permafrost area gives $\widetilde{D}_{tot}$ of $\sim$5.5 $\pm$0.5 x $10^3$ km$^3$. Assuming $z_{max}$ is 2 m, $\widetilde{F}_{tot}$ is then $\sim$22.5 $\pm$0.5 x $10^3$ km$^3$.

## 3  Results

In a global climate model the permafrost dynamics is affected by both the driving climate and by the paramterisations used to translate the meteorology into the presence or absence of permafrost namely the land surface module. Here we separate out these two factors and where possible identify the relative uncertainties introduced.





## 3.1 Driving climate

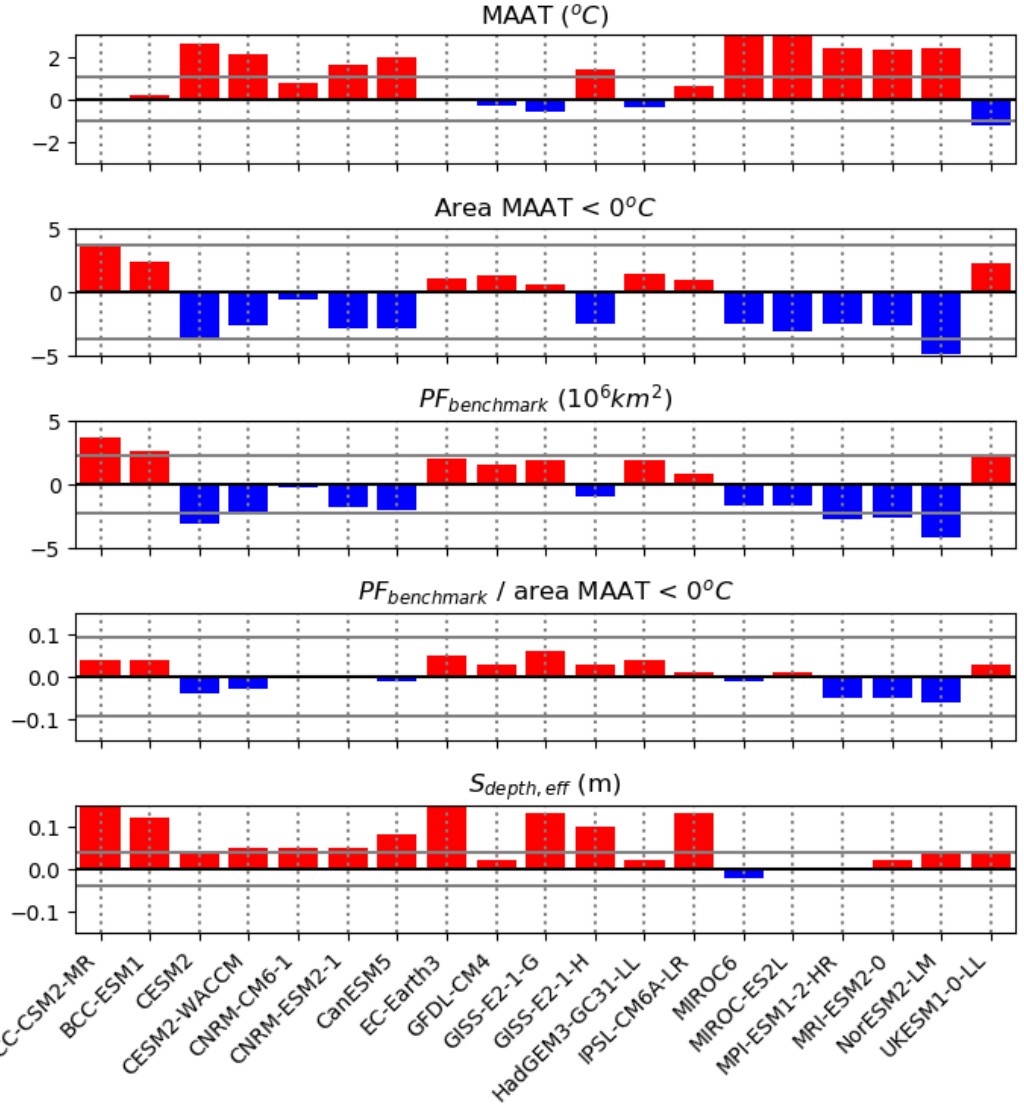

**Figure 2.** The driving climate characteristics of the CMIP6 multi-model ensemble compared with the observations for the period 1995–2014. The red bars are where the model value is greater than the observations and the blue bars are where the model value is less than the observations. $S_{depth,eff}$ is for the period 1998-2016 and is not available for every ensemble member.

Figure 2 shows the differences between the observations and the CMIP6 models for relevant climate-related characteristics of the permafrost affected region ($PF_{aff}$) defined by the CCI-PF data (Table 2). The horizontal grey lines on Figure 2 represent 230 ±15% of the observed value. Absolute values for individual models and the observations are given in Table S1.1. These can



also be compared with the CMIP5 multi-model ensemble (Figure S2.1 and Table S2.2).

The $MAAT$ is, to first order, the driver of the presence or absence of permafrost (Chadburn et al., 2017). The ability of the climate models to correctly simulate the northern high latitudes $MAAT$ is assessed in the top two panels of Figure 2.

The majority of the CMIP6 models are warmer than the observations and have too small an area where the land surface is less than 0°C. In general the models fall within $\pm2$°C of the observed $MAAT$ (-6.8°C) and within $\pm4.0$ x $10^6$km$^2$ of the observed area where $MAAT$ is less than 0°C (24.4 x $10^6$km$^2$). However any differences will cause biases in any subsequent estimate of permafrost presence. These can be quantified by comparing $PF_{benchmark}$ with $PF_{ex}$ derived using the WFDEI air temperatures and Chadburn et al. (2017) – 15.1 x $10^6$km$^2$. Figure 2 shows, as might be expected, that the $PF_{benchmark}$ is

underestimated by the majority of models with a range of extents between 11.0 and 18.7 x $10^6$km$^2$ (Table S1.2). The majority of models have a $PF_{benchmark}$ greater than the 13.5 x $10^6$km$^2$ from the CCI-PF data. The differences between models appear smaller when $PF_{benchmark}$ is normalised by the area where $MAAT$ is less than 0°C. These range between 0.58 and 0.68 (Table S1.2). The observations have a value of 0.62 when using the Chadburn et al. (2017) relationship and WFDEI. These differences between models are caused by differences in the latitudinal dependence of $MAAT$ for temperatures between 0

and -7.6°C – the threshold temperatures of permafrost presence/absence and continuous permafrost respectively. The left hand plot in Figure 3 shows the mean probability of $PF_{benchmark}$ for the CMIP6 multi-model ensemble. Any region where there is permafrost using this definition is shaded in purple. Superimposed is the contour plot of probability of permafrost from Obu et al. (2019) with the orange lines the limits of 50% permafrost. In general the continuous permafrost area is well represented as 100% permafrost meaning that all of the models can represent the area of continuous permafrost. However, the permafrost

extends further south in a small handful of models, as might be expected from the spread in Figure 2. Figures for individual models are shown in Figure S1.1. The $PF_{benchmark}$ for each model can be used as the reference data for evaluating the ability of the land surface component to appropriately estimate permafrost presence.



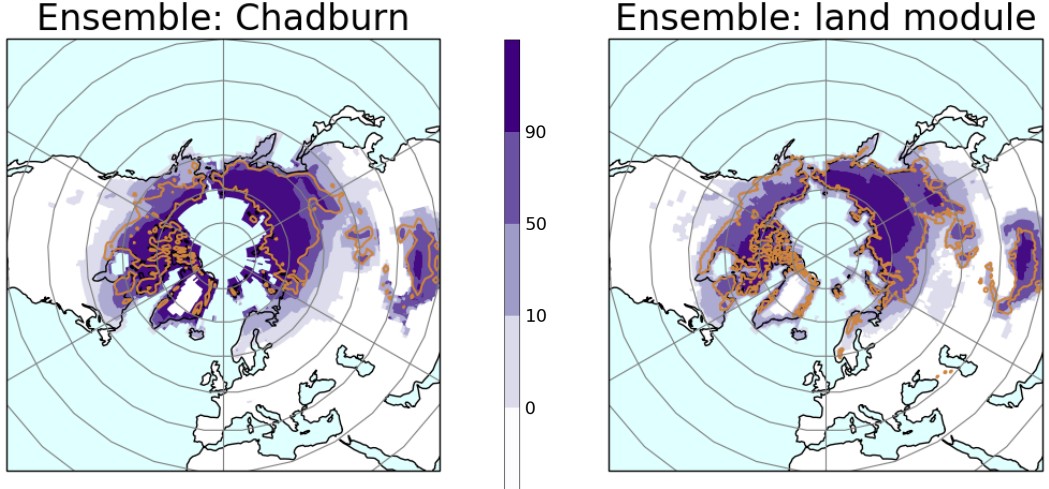

**Figure 3.** The left hand figure shows the ensemble probability of permafrost using Chadburn et al. (2017) relationship for each model ($PF_{benchmark}$). The right hand figure shows the ensemble probability of permafrost where permafrost is defined by the temperature at the $D_{zaa}$ or the lowest model level for the models with the shallower soil profile ($PF_{ex}$). These plots are the mean for 1995–2014. The orange lines are the limits for 50% permafrost from the CCI-PF data (Obu et al., 2019).

The CMIP6 multi-model ensemble can be compared with the CMIP5 multi-model ensemble (Table 3). The standard time periods are slightly different for each model ensemble: the CMIP6 climatologies are for 1995–2014 the the CMIP5 climatologies are for 1986-2005. Therefore the observed values (except for $S_{eff}$ which covers a more limited time period) are slightly different with the $MAAT$ for CMIP6 about $0.3°C$ warmer than for CMIP5 and the area where the land surface is less than $0°C$ is 0.4 x $10^6$km$^2$ larger for CMIP5. Overall the two model ensemble means agree with the observations for the metrics derived from air temperature with the majority of the constituent models falling within $\pm15\%$ of the observed values. Table 3 shows the CMIP6 models are slightly warmer than the observations and the CMIP5 models are slightly colder.





| | CMIP6 | | | CMIP5 | | |
|---|---|---|---|---|---|---|
| | observations (1995-2014) | model ens. mean - obs. | Percentage within ±15% | observations (1986-2005) | model ens. mean - obs. | Percentage within ±15% |
| $MAAT$ (°C) | -6.8 | 0.44 | 47 | -7.1 | -0.13 | 47 |
| area $MAAT$<0°C (x $10^6$km$^2$) | 24.4 | -0.88 | 89 | 24.8 | 0.46 | 84 |
| $PF_{benchmark}$ (x $10^6$km$^2$) | 15.1 | -0.35 | 68 | 15.7 | 0.65 | 73 |
| $PF_{benchmark}$/area $MAAT$<0°C | 0.62 | -0.0 | 100 | 0.61 | 0.03 | 94 |
| $S_{depth,eff}$ (m) | 0.25 | 0.07 | 21 | 0.25 | 0.07 | 21 |

**Table 3.** A summary of the CMIP6 climate evaluation metrics compared with both the observations and CMIP5. Where relevant, statistics are given for $PF_{aff}$ defined by CCI-PF 2. The difference between the ensemble mean and the observations are shown plus the percentage of the model ensemble within ±15% of the observations. It should be noted that the observed $S_{depth,eff}$ is for the period 1998-2016.

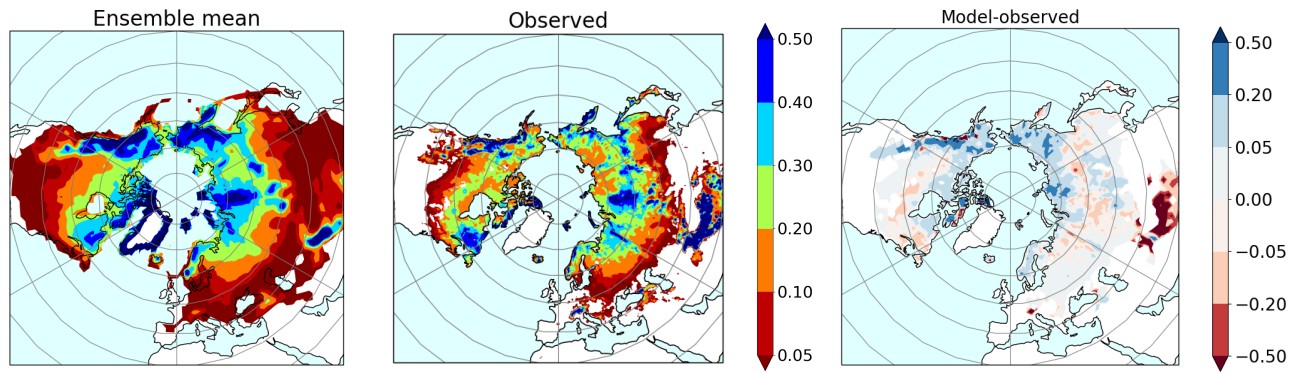

**Figure 4.** $S_{depth,eff}$ for 1999-2014 for the ensemble mean of the CMIP6 models compared with the CMC observations. The right hand plot shows the differences between the model ensemble mean and the observations.

The precipitation will also affect the presence or absence of permafrost - in particular any snow lying on the ground will insulate the soil. The land surface scheme translates snowfall to snow lying on the surface and quantifies its insulating capacity. Therefore biases in both the snow amount and snow physics will influence the snow insulation. The snow amount can be
represented by the $S_{depth,eff}$. Figure 4 shows the model ensemble median $S_{depth,eff}$ compared with the observations from the CMC snow depth analysis (Brown and Brasnett, 2010) for the time period 1998–2014 and Figure S1.2 shows $S_{depth,eff}$ for the individual models. All grid cells with $S_{depth,eff}$ less than 2 cm are masked. $S_{depth,eff}$ for the arctic is generally greater than 0.2 m in the model ensemble mean. The observations have some regions at the northern tundra where the effective snow depth is slightly shallower than 0.2 m which are not reflected on the ensemble mean nor in the individual models. This results in
a tendency for the models to slightly over-estimate the snow depth in the tundra and slightly under estimate in the boreal region. The snow region extends further south in the ensemble mean than in the observations which reflects some of the variability





between models. Individual CMIP6 models (Figure S1.2) that notably over-estimate the $S_{depth,eff}$ include EC-Earth3, BCC-CSM2-MR, and GISS-E2-1-G. Only 21 % of the models have a mean $S_{depth,eff}$ within $\pm15\%$ of the observations Table 3. It should be noted that $S_{depth,eff}$ will be moderated by snow physics and in the case of snow the climate biases cannot be cleanly

separated from the land surface biases.

## 3.2   Land surface module

The land surface modules translate the driving climate into the permafrost dynamics. In effect they quantify the offsets shown in Figure 1. Figure 5 shows the spread of these offsets as a function of $MAAT$ for the CMIP6 multi-model ensemble along with an estimate of the observed surface and thermal offsets. This spread was calculated independently for each model by

binning the offsets into 0.5°C bins and calculating the median value of each bin. The winter offset is by far the largest offset with the largest uncertainty and it is strongly dependent on $MAAT$. Therefore snow plays a dominant role in the relationship between $MAGT$ and $MAAT$. The summer and thermal modelled offsets both have a small negative value, cover a smaller range of values, and are only slightly dependent on $MAAT$. In comparison to the observations and assuming the summer offset is small, the model-simulated winter offsets are possibly slightly to small at the warmer temperatures and slightly too large at

the colder temperatures. Figure S1.3 shows the variation between models is relatively large between the individual models. For example, MPI-ESM1-2-HR has very little difference between the $MAAT$ and the $MAGT$ with all offsets of the order 1°C or less whereas UKESM1-0-LL has a winter offset which reaches over 10°C in very cold conditions. However, comparing the CMIP5 model ensemble members with the CMIP6 model ensemble (Figures S1.3 and S2.5 suggests that there is a general improvement since CMIP5 when compared with the observations.






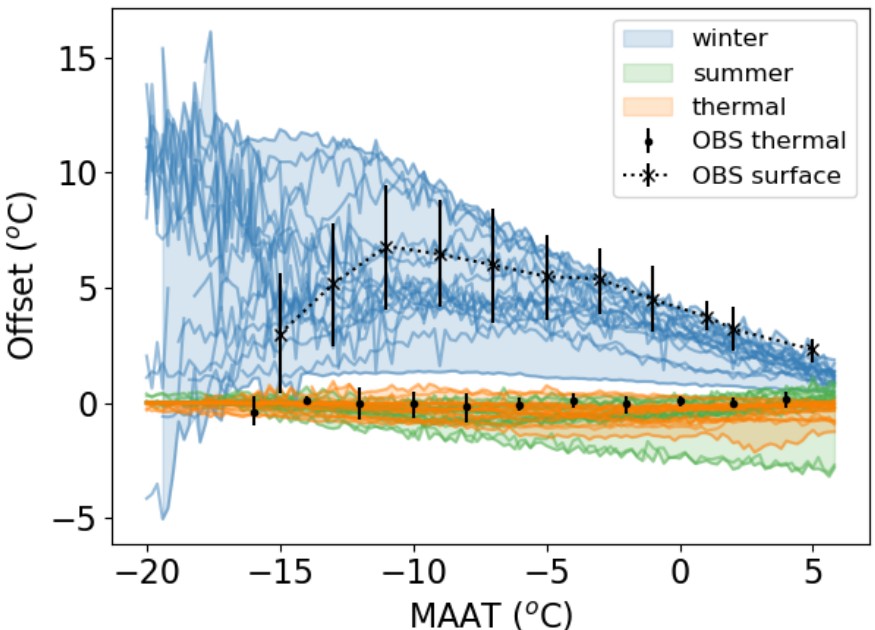

**Figure 5.** Ensemble spread of the median winter, median summer and median thermal offsets for the CMIP6 model ensemble. Individual models for the CMIP6 ensemble are shown as lines and identified in Figure S1.3 and for the CMIP5 ensemble in Figure S2.5. The observed surface and thermal offsets summarised from the available point data (Zhang et al., 2018) are added in black.

### 3.2.1    Mean annual ground temperature

Figure 5 suggests that in order for a land surface module to be able to accurately represent permafrost it needs to be able to represent the insulating ability of the lying snow. This is assessed in Figure 6 which shows the insulating capacity of the snow in terms of the difference between the winter air temperature and winter 20 cm soil temperature. The offsets are a function of

both $MAAT$ (not shown) and $S_{depth,eff}$ with the lower the $MAAT$ and higher $S_{depth,eff}$ the bigger the offset Wang et al. (2016). The available models reflect the general increase in offset with increasing $S_{depth,eff}$ for the shallow snow and the saturation of this relationship for the deepest $S_{depth,eff}$ to varying degrees of accuracy. Both CNRM-CM6-1 and CMRM-ESM2-1 have offsets which are too small at all values of $S_{depth,eff}$ and show no signs of saturating at higher values of $S_{depth,eff}$. Generally the models have too low an offset. Notable exceptions include HadGEM3-GC31-LL/UKESM1-0-LL which has too

much insulation for deeper snow and and BCC-CSM2-MR/BCC-ESM1 which has relative good insualtion for the deeper snow depths but still not quite enough for the shallower snow depths.

The impact of the snow offset needs to be interpreted in combination with the $S_{depth,eff}$ in order to evaluate the impact of the snow insulation on permafrost dynamics. Figure 4 shows that arctic snow depths are relatively shallow. Therefore, because



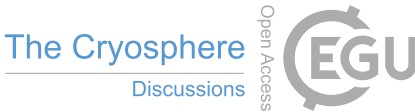

there is a non-linear relationship between offset and $S_{depth,eff}$ small differences in $S_{depth,eff}$ will have a big impact on the insulating ability. The models tend to slightly over-estimate $S_{depth,eff}$. This means that in the models where the relationship between offset, $MAAT$ and $S_{depth,eff}$ is good there is likely to be too much snow insulation and in models where the offset is too small the overall snow insulation might be better than expected. This will affect the ability of the models to simulate the $MAGT$, particularly in the high arctic where the winter temperatures are very cold. In these regions the permafrost is

likely to be continuous. In regions of discontinuous and sporadic permafrost the presence or absence of permafrost will be strongly influenced by the snow insulation. In these warmer areas the relationship between winter offset and $S_{depth,eff}$ is well simulated by nearly all of models.

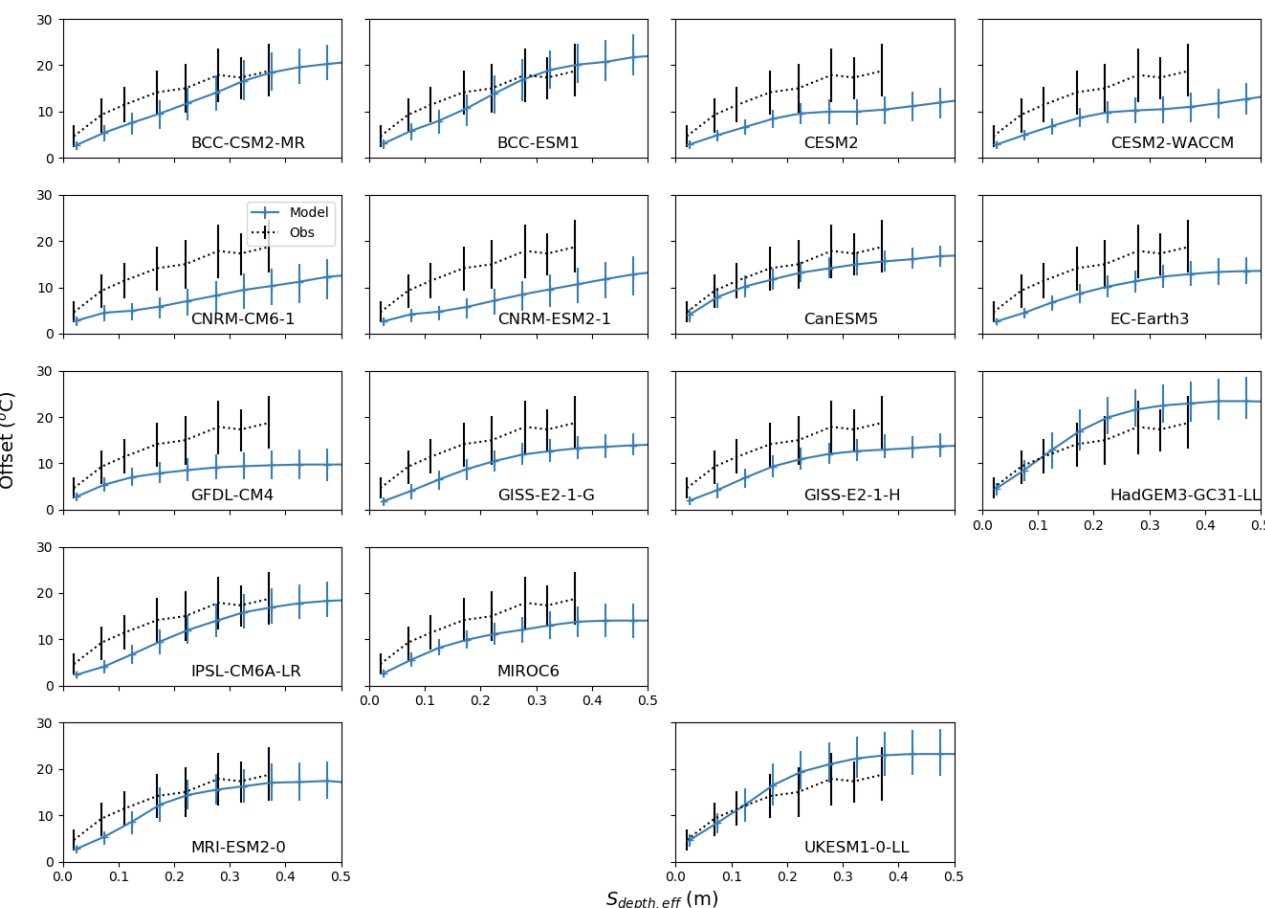

**Figure 6.** Differences between the $MAAT$ and the annual mean soil temperature at 0.2 m for the winter as a function of $S_{depth,eff}$. The climatological period of 1995–2014 is shown for the CMIP6 models. The blue points with the error bars are the model data and the dotted black lines and errorbars are the observations derived using the data from Zhang et al. (2018).





Figure S2.6 shows the equivalent plots for the available CMIP5 models. A similar pattern is observed where the models underestimate the snow insulation. Although limited availability means it is hard to compare individual models between the
CMIP5 and CMIP6 ensemble, specific models can be identified. Specifically CanESM and MIROC show improvements; MRI, GISS and BCC show no change and CESM shows some degradation.

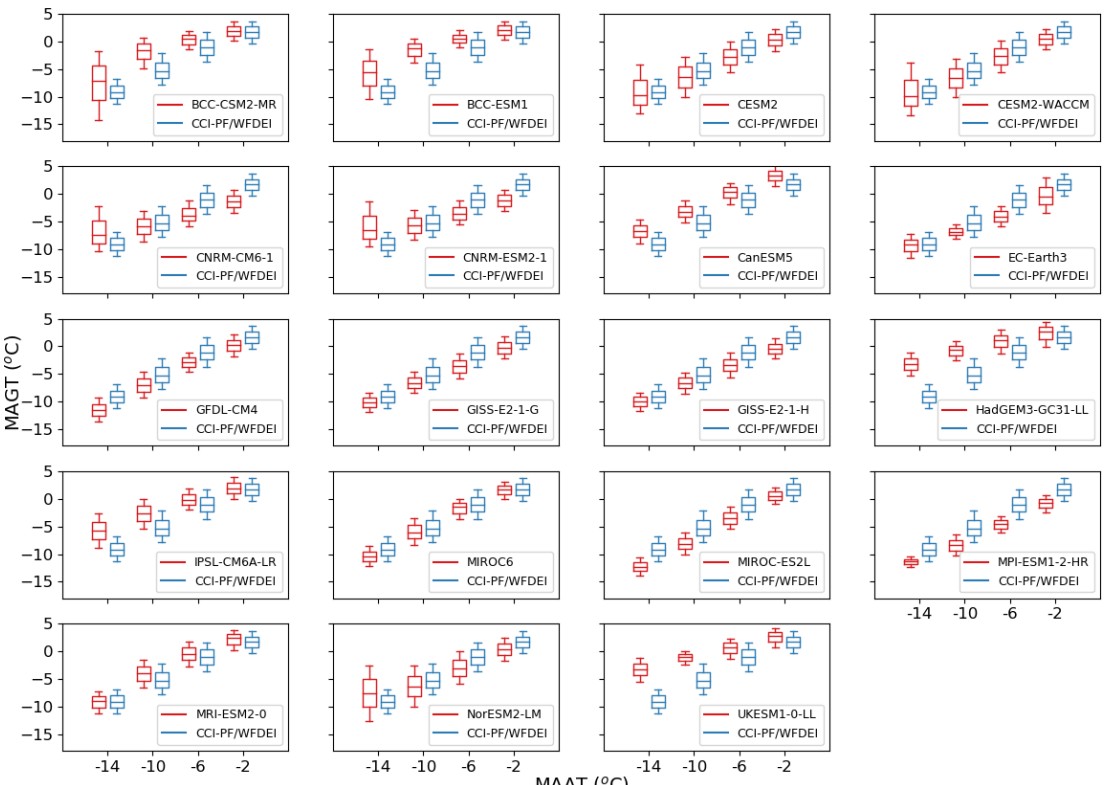

**Figure 7.** $MAGT$ as a function of local $MAAT$ for the CMIP6 models and the climatological period 1995–2014. The $MAGT$ observations were taken from the CCI-PF data set and the $MAAT$ from the WFDEI data.

Figure 7 shows the combined impact of the three offsets on the relationship between $MAGT$ and $MAAT$ compared with an observational-based assessment made using the CCI-PF $MAGT$ and the WFDEI $MAAT$. As expected the $MAGT$ increases
with $MAAT$. In addition the CCI-PF $MAGT$ is approximately 4.5 °C warmer than the WFDEI $MAAT$. As discussed earlier this difference is dominated by the winter offset but the summer and thermal offsets also contribute. At the warmer temperatures there is a small reduction in this difference because the winter offset is smaller at warmer temperatures (Figure 6). Also shown





are the same relationships for the models. The differences in snow insulation discussed above are also reflected here. For example, the BCC and MOHC models have a much larger difference between $MAAT$ and $MAGT$ than the observations at
the colder temperature because there is too much snow on the ground in the high arctic. The MIROC6 model has a very similar relationship between $MAAT$ and $MAGT$ as the observations despite not being able to recreate the large snow offset at the very low temperatures. CESM has a similar relationship compared with the observations despite not recreating the relationship between snow offset and $MAAT$ in Figure 6, in this case the biases are cancelling out. A comparison with the CMIP5 multi-model ensemble (Figure S2.7 shows similar differences. It should be noted that this CCI-PF estimate of $MAGT$ is a model
derived reanalysis and the uncertainties are likely under-estimated in the current analysis.





### 3.2.2 Active layer thickness

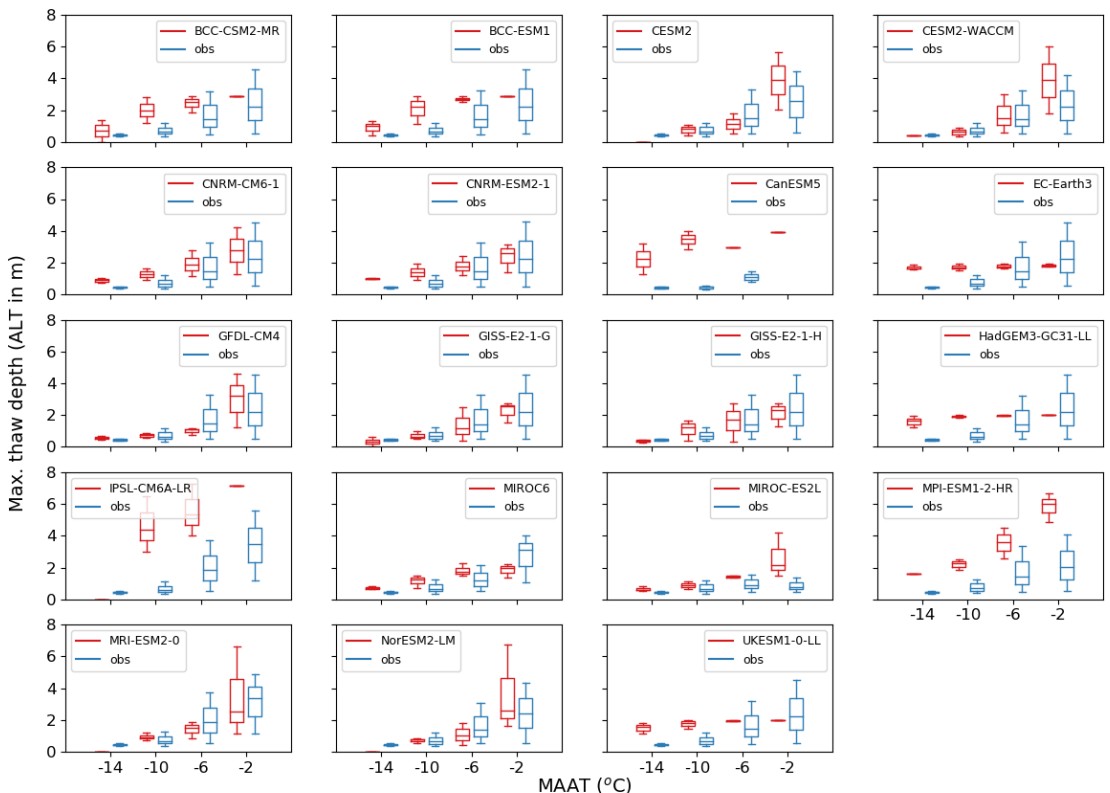

**Figure 8.** Active layer thickness ($ALT$) as a function of local $MAAT$ for the CMIP6 models and the climatological period 1995–2014. Observations of active layer are from the CALM sites and the air temperatures are from the large scale WFDEI data set. Only sites where the specified model simulates permafrost are included in this analysis.

An assessment of the thaw depth of the permafrost during the summer gives another indication of whether the model has the correct physics. Differences between model and observations are apparent in Figure 8. The observed relationship between the active layer thickness and the $MAAT$ in the CALM data set is shown in blue. Both the observed active layer and the spread of possible active layer depths increase within increasing $MAAT$. The spread of values increases because the active layers are more strongly impacted by environmental factors other than air temperature such as topography, soil type and solar radiation at the warmer temperatures. For each model only the observed CALM sites where there is model simulated permafrost are included in the assessment so the observations are slightly different depending on model. The maximum observed active layer





thickness is about 4.5 m. Any models with a maximum soil depth less than 4.5 m will be unable to represent this value (Table
1). This is the case for GISS where the depth of the middle layer is 2.7 m and the $ALT$ is constrained to 2.7m at the warmer
temperatures. Poor vertical discretisation of the soil such as the 3 layers in CanESM5 can introduce large variability into the
derivation of $ALT$. Although the average number of soil layers and the average soil depth increases between the CMIP5 and
CMIP6 ensembles (CMIP6: Table 1 and CMIP5: Table S2.1) this is not universally true.

345        About half of the models have relationship between $ALT$ and $MAAT$ which are comparable with the observations. These
are mainly the models with the deeper soils. Other models have very deep $ALT$ for example, MPI-ESM1-2-HR and IPSL-
CM6A-LR. These both have sufficiently deep soil profiles but thaw too quickly in the summer likely because these models
don't represent the latent heat of the water phase change. EC-Earth, UKESM1-0-LL and HadGEM3-GC31-LL have active
layers around 2m irrespective of $MAAT$. In the case of the MOHC models this is worse than the CMIP5 version of the model
where $ALT$ was dependent on $MAAT$, just with a smaller uncertainty range (Figure S2.8). This is because there is a large
increase in the $MAGT$ between the CMIP versions ($MAGT$ is -7.0 °C and -5.8 °C for CMIP5 and $MAGT$ is -0.3 °C and
-0.9 °C for CMIP6) caused by adding a multilayered snow scheme (Walters et al., 2019). The inclusion of organic soils and the
addition of a moss layer improves the insulating capacity and ability of the soil to hold water and will reduce this thaw depth
(Chadburn et al., 2015).

**3.2.3   Permafrost extent, annual thawed volume ($\widetilde{D}_{tot}$) and annual frozen volume ($\widetilde{F}_{tot}$)**

$MAGT$ and $ALT$ and their relationships with $MAAT$ are important diagnostics for model physics. However, permafrost ex-
tent, annual thawed and frozen volume are more relevant when exploring the impacts of changing permafrost dynamics under
future climate change. There are observational-based estimates of permafrost extent (Brown et al., 2003; Obu et al., 2019;
Chadburn et al., 2017) available for evaluation and this paper introduces an observational-based estimate of annual thawed and
frozen volume by extrapolating the site specific relationship between annual mean thawed fraction ($\widetilde{D}$) and $MAGT$ to the
larger scale using the CCI-PF data set.

Figure 9 shows the relationship between annual mean thawed fraction ($\widetilde{D}$) and $MAGT$ for the available CALM and GTNP
data sets is relatively well constrained. As expected $\widetilde{D}$ increases with increasing $MAGT$. Figure 9 also shows the ability of
the models to replicate this relationship. At the warmer temperatures the models tend to show much more variability than the
observations. In models such as EC-Earth3, UKESM1-0-LL, HadGEM3-GC1-LL this is likely reflecting the sensitivity to the
duration over which the soil is thawed - because in these models the $ALT$ is very similar at all temperatures (Figure 8). Overall
the models follow a similar trend of increasing $\widetilde{D}$ with increasing $MAGT$. There are notable discrepancies for IPSL-CM6A-
LR and MPI-ESM1-2-HR which might be expected because these two models simulate very deep $ALT$. $\widetilde{D}$ can be converted
to annual thawed and frozen volumes ($\widetilde{D}_{tot}$ and $\widetilde{F}_{tot}$) using the monthly profiles of modelled soil temperature and compared
with the observational-based estimate of $\widetilde{D}_{tot}$ and $\widetilde{F}_{tot}$ discussed in Section 2.3.4. Figure 10 summarises $PF_{ex}$, $\widetilde{D}_{tot}$ and $\widetilde{F}_{tot}$





for each of the CMIP6 models.

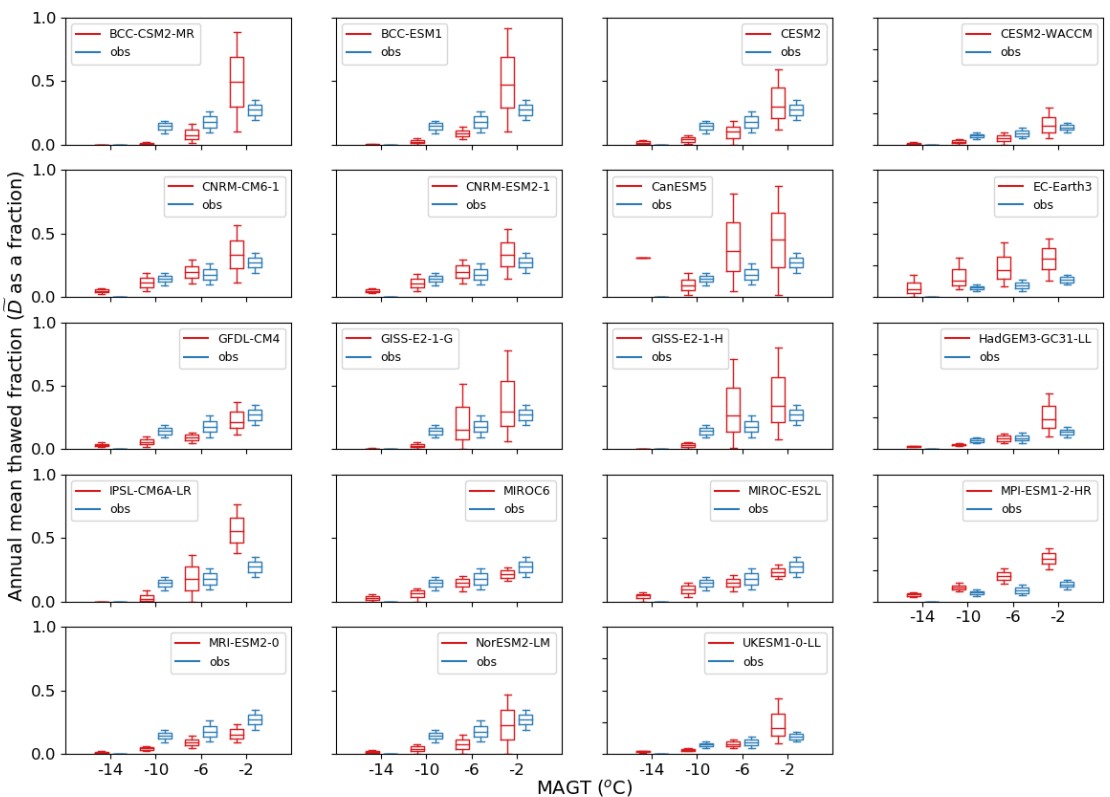

**Figure 9.** Relationship between the annual mean thawed fraction ($\widetilde{D}$) and the $MAGT$ from the site observations and the models in CMIP6 for the climatological period 1995–2014.

$PF_{ex}$ is derived using the temperature at $D_{zaa}$ or the lowest model level if the soil profile is too shallow. [The spatial
distribution for individual CMIP6 models are shown in Figure S1.4]. In Figure 10, the models where the permafrost is defined at $D_{zaa}$ have black hatching. The bars with the black outlines show $PF_{benchmark}$ and the grey shaded area shows the range of the observed data sets. A comparison of $PF_{benchmark}$ with the observations shows that the models fall close to these observed values in all cases as is expected from Table S1.2. $PF_{ex}$ should be compared with $PF_{benchmark}$ and not the observations and any differences between the two quantifies the bias added by the land surface model. Any differences here are again
dominated by the snow insulation which is a combination of the $S_{depth,eff}$ and the relationship between $MAGST$, $MAAT$ and $S_{depth,eff}$. When compared with the CMIP5 multi-model ensemble (Figure S2.10), the spread of values is smaller with





the differences between $PF_{ex}$ and $PF_{benchmark}$ significantly lower in some cases. Assessing Figure 10 in conjunction with Figure S1.2 and Figure 6, suggests some improvement in snow insulation between the CMIP5 multi-model ensemble and the CMIP6 multi-model ensemble.

Figure 10 also shows $\widetilde{D}_{tot}$ and $\widetilde{F}_{tot}$ for each CMIP6 model. These metrics are calculated for the present-day permafrost region defined by the model specific $PF_{ex}$ and they sum to $PF_{ex}$ multiplied by 2 m. Any variability is caused by a combination of differences in the $MAGT$ and variations in the relationships between $\widetilde{D}$ / $MAGT$ / $MAAT$. These variations leads to the considerable variation in $\widetilde{D}_{tot}$ and $\widetilde{F}_{tot}$ shown in Figure 10. There is a tendency for the models to over-estimate $\widetilde{D}_{tot}$ and under-estimate $\widetilde{F}_{tot}$. This arises partly because of the large variability in $\widetilde{D}$ at the warmer air temperatures i.e. in the discontinuous

and sporadic permafrost (Figure 9). The CMIP6 multi-model ensemble has a similar uncertainty to that of CMIP5 suggesting little improvement in the quantification of $\widetilde{D}_{tot}$ and $\widetilde{F}_{tot}$ between ensembles.

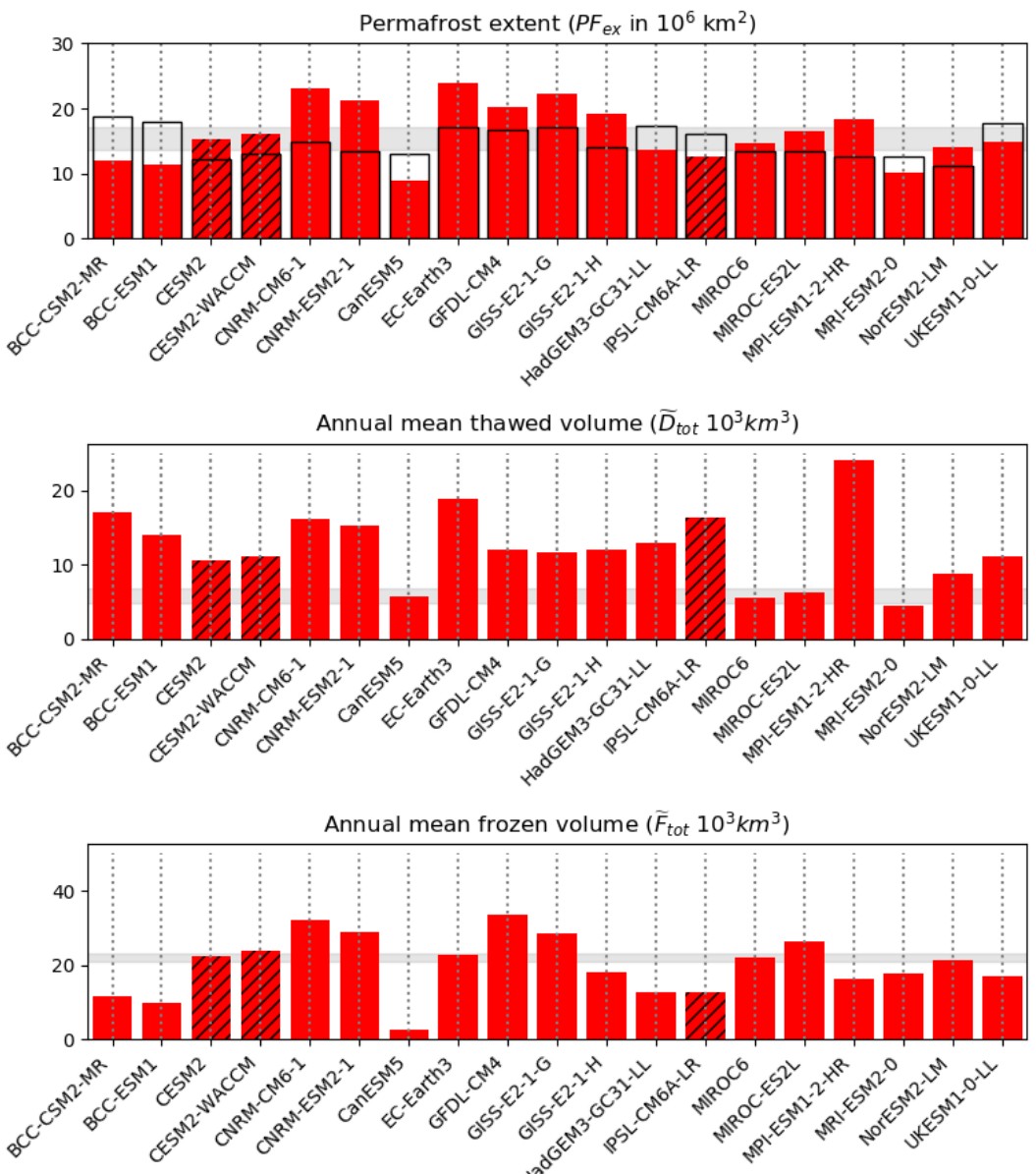

**Figure 10.** (a) Permafrost extent ($PF_{ex}$) derived using the mean temperature at $D_{zaa}$ or the mean temperature at the bottom of the soil layer if the soil profile is too shallow 1 is shaded in red. The models where the permafrost is defined using $D_{zaa}$ have black hatching. The empty bars with black outlines are the $PF_{benchmark}$ derived from the Chadburn et al. (2017) relationship. The grey shaded area is the range expected from observations. All the results are for the CMIP6 models for the climatological period of 1995–2014. The middle figure shows the annual thawed volume ($\widetilde{D}_{tot}$) of the top 2 m for the model-specified permafrost region and the bottom figure shows the annual mean frozen volume of the top 2 m ($\widetilde{F}_{tot}$) for the model-specified permafrost region. These two should add to the permafrost extent shown in (a) multiplied by 2 m.





|  | H | | | | |
|---|---|---|---|---|---|
|  | **CMIP6** | | | **CMIP5** | |
|  | observed (range) | ensemble mean ($25^{th}$-$75^{th}$ percentile) | Percent within obs. range | ensemble mean ($25^{th}$-$75^{th}$ percentile) | Percent within obs. range |
| $PF_{ex}$/area $MAAT$<0°C | 0.62 (0.55 to 0.77) | 0.55 (0.46 to 0.72) | 36 | 0.56 (0.34 to 0.81) | 15 |
| $\widetilde{D}_{tot}$/area $MAAT$<0°C | 0.22 (0.20 to 0.25) | 0.37 (0.23 to 0.47) | 26 | 0.31 (0.19 to 0.44) | 21 |
| surface offset (°C; -14°C< $MAAT$ <-2°C) | 5.7 (4.2 to 7.1) | 4.5 (3.4 to 5.0) | 26 | 3.9 (1.5 to 6.5) | 15 |
| thermal offset (°C; -14°C< $MAAT$ <-2°C) | 0.03 (-0.15 to 0.15) | -0.34 (-0.55 to -0.10) | 26 | -0.16 (-0.35 to -0.10) | 26 |
| $MAGT$/$MAAT$ | 0.91 (0.88 to 0.93) | 0.78 (0.59 to 0.96) | 26 | 0.85 (0.74 to 0.95) | 31 |
| $ALT$ (m; -12°C< $MAAT$ <-10°C) | 0.5 (0.4 to 0.8) | 1.52 (0.7 to 2.0) | 36 | 1.5 (1.0 to 2.0) | 15 |
| $ALT$ (m; -6°C< $MAAT$ <-4°C) | 1.2 (0.6 to 2.0) | 2.77 (1.7 to 2.9) | 42 | 2.8 (1.8 to 2.9) | 36 |

**Table 4.** CMIP6 model evaluation metrics summary compared with observations and CMIP5. Individual CMIP6 models are in Table S1.2 and Table S2.3.

### 3.2.4 Evaluation metrics

This section summarises some basic evaluation metrics which can be applied to the models to quantify the ability of the land surface module to represent permafrost dynamics. Relevant climate-related metrics are summarised in Table 3 and shown in

Table S1.1 for individual CMIP6 models and Table S2.2 for individual CMIP5 models. Table 4 shows a summary of the land-related evaluation metrics for the observations and the two different model ensembles along with the percentage of each ensemble falling within the range with individual models shown in Table S1.2 and Table S2.3. These metrics should be relatively independent of climate and reflect the behaviour of the land surface module.There is better agreement with the observations for the climate-related metrics than for the land-surface related metrics within only a very small number of models falling within

the range of the observations for the land surface components. In addition, there is no consistency in model performance - no model performs well for every evaluation metric (Table S1.2 and Table S2.3). Overall, the percentage of the models which fall within the observed range is relatively low with the majority falling outside the range of the observations (Table 4). However, there is a very small improvement in the percentage of models that fall within the observed range between the CMIP5 ensemble and the CMIP6 ensemble.




### 3.3 Future projections

This paper makes future projections of $PF_{benchmark}$, $PF_{ex}$, $\widetilde{D}_{tot}$ and $\widetilde{F}_{tot}$ as a function of global mean temperature change ($GMT$). The results are shown in Figure 11 for all of the available SSP scenarios. $PF_{benchmark}$ depends solely on the projections of air temperature; $PF_{ex}$ depends on the annual mean soil temperature at depth; and $\widetilde{D}_{tot}$ and $\widetilde{F}_{tot}$ depend additionally on

the thawed component of the soil. We assume the results are scenario independent and calculate the mean of each diagnostics for each model after binning into 0.1°C global mean temperature bins. Anomalies are then calculated with respect to the values where the $GMT$ change is 0.0 °C. Data are excluded any time the simulated permafrost extent falls below 1.0 x $10^6$km$^2$. For most of the models these relationships are approximately linear for temperature changes up to around 3 °C.

The sensitivity of $PF_{benchmark}$ to increasing $GMT$ shows a loss of between 3.3 and 4.1 x $10^6$ km$^2$/°C ($25^{th}$ to $75^{th}$ percentile; Table 5) in the CMIP6 multi-model ensemble. This derivation of $PF_{benchmark}$ uses the observed relationship between $MAAT$ and the probability of permafrost Chadburn et al. (2017) and assumes it is for an equilibrium state. Therefore this sensitivity is highly dependent on the arctic amplification in the models. The variability in $PF_{benchmark}$ between the different models and different model ensembles is relatively small with no obvious outliers. The sensitivities fall to the lower end of the

equilibrium sensitivity proposed by Chadburn et al. (2017) who present a loss of $4.0^{+1.0}_{-1.1}$ x $10^6$ km$^2$/°C.

The sensitivity of $PF_{ex}$ to increasing $GMT$ is related to both the climate and the land surface module and has a wider range of values - 1.8 to 3.0 x $10^6$ km$^2$/°C for the $25^{th}$ to $75^{th}$ percentile; Table 5. This is around 12 to 20 % /°C of the present day permafrost and is indistinguishable from the CMIP5 sensitivities. The loss of permafrost derived from $PF_{ex}$ is less than from

$PF_{benchmark}$. One reason for this is that $PF_{ex}$ represents a transient response to $GMT$. Despite the shallow soil profile in the majority of the models, the methodology used to derive the $PF_{ex}$ means there will be some implicit time delay in the heat transfer from the surface. There are a few outliers in Figure 11b which have very different sensitivities of $PF_{ex}$ to increasing $GMT$. These outliers include the MIROC models which project a low sensitivity of $PF_{ex}$ to $GMT$ in both the CMIP6 and CMIP5 ensembles (see also Figure S2.11). Although there is some improvement in the winter offset in MIROC between

CMIP5 and CMIP6 (Figure S2.6 and 6), there is still too little snow insulation in the present day model because $S_{depth,eff}$ is relatively shallow. In addition the slope of the relationship between $MAGT$ and $MAAT$ is greater than 1. These factors mean MIROC has a 'permafrost-prone climate' (Slater and Lawrence (2013)).

The increase in mean thawed volume ($\widetilde{D}_{tot}$)/decrease in mean frozen volume ($\widetilde{F}_{tot}$) is actually relatively consistent between

the different models and ranges from 4.3 to 5.5 x $10^3$ km$^3$ / °C ($25^{th}$ to $75^{th}$ percentile; Table 5). This represents a mean change of state in around 20-30% of the permafrost in the top 2m of the soil per degree increase in $GMT$. The MIROC models are not outliers - the sensitivity of $\widetilde{D}_{tot}$ and $\widetilde{F}_{tot}$ to $GMT$ in MIROC is falls within the spread of the other models and the relationship between $\widetilde{D}$ and $MAGT$ is comparable with the observed relationship. This suggests that the summer thawing processes are well represented in the MIROC models despite some biases in the sensitivity of the mean deep soil temperature





to temperature change. In contrast, the IPSL-CM6A-LR model has a much lower sensitivity of $\widetilde{D}_{tot}$ and $\widetilde{F}_{tot}$ to $GMT$. This

is because it doesn't represent the latent heat required for thawing and therefore has too deep an active layer. However, it falls

within the spread of the sensitivity of the $PF_{ex}$ to $GMT$. This is mainly because it has a very deep soil profile and the $PF_{ex}$

is diagnosed at $D_{zaa}$. These examples illustrate how the parameterisation of permafrost physics can affect the projections in

different ways.








**Figure 11.** Projections of (a) loss of permafrost extent defined as $PF_{benchmark}$ derived from the $MAAT$; (b) loss of permafrost extent defined as $PF_{ex}$ derived from the soil temperatures; (c) increase in annual mean thawed volume; and (d) loss of annual mean frozen volume as a function of global mean temperature change for the CMIP6 models. All the available scenarios are superimposed on one figure and the results binned into $0.1°C$ global mean temperature change ($GMT$) bins. $PF_ex$ is greater than $1 \times 10^6$ km$^2$ in all cases.



|  | CMIP6 | | | | | CMIP5 | | | | |
|---|---|---|---|---|---|---|---|---|---|---|
| Percentile (%) | 5 | 25 | **50** | 75 | 90 | 5 | 25 | **50** | 75 | 90 |
| $PF_{benchmark}$/$GMT$ ($10^6$ km$^2$/$^oC$) | -4.6 | -4.1 | **-3.4** | -3.3 | -2.7 | -4.2 | -3.8 | **-3.5** | -3.1 | -2.9 |
| $PF_{ex}$/$GMT$ ($10^6$ km$^2$/$^oC$) | -3.7 | -3.0 | **-2.4** | -1.8 | -0.3 | -3.5 | -3.2 | **-2.4** | -1.8 | -0.7 |
| $\widetilde{D}_{tot}$/$GMT$ ($10^3$ km$^3$/$^oC$) | 2.8 | 4.3 | **5.0** | 5.5 | 6.7 | 3.0 | 4.3 | **4.7** | 5.6 | 6.4 |

**Table 5.** Projections of loss of $PF_{benchmark}$, $PF_{ex}$ and $\widetilde{D}_{tot}$ as a function of sensitivity to global mean temperature change ($GMT$).

## 4 Discussion and Conclusion

This paper examines the permafrost dynamics in both the CMIP6 multi-model ensemble and CMIP5 multi-model ensemble using a wide range of metrics. As far as possible the metrics were defined so as to identify the effect of biases in the climate separately to biases in the land surface module. Overall, the two model ensembles are very similar in terms of climate, snow

and permafrost physics and projected changes under future climate change. This paper does not attempt to document specific improvements to individual models in any detail, in fact the CMIP5 and CMIP6 ensembles contain slightly different models. However, it is apparent that the snow insulation is improved in a few of the models which results in overall less variability in the permafrost extent ($PF_{ex}$) in the CMIP6 ensemble than in the CMIP5 ensemble. In general, the ability of the models to simulate of summer thaw depths is not improved between ensembles. One reason for this remains limitations caused by

shallow and poorly resolved soil profiles.

Over the past few years there has been a lot of model developments improving the representation of northern high latitude processes in land surface models (e.g. Chadburn et al. (2015); Porada et al. (2016); Guimberteau et al. (2018); Cuntz and Haverd (2018); Lee et al. (2014); Burke et al. (2017a); Hagemann et al. (2016); Lee et al. (2014) but many of these remain to be

included within the CMIP6 model versions. In particular, excess ground ice which exists as ice lenses or wedges in permafrost soils is a key process that is not included in the current generation of CMIP models. This means that thermokarst processes whereby ice-rich permafrost ground thaws and collapses are not represented. Thermokarst landscapes cover about 20 % of the northern permafrost region (Olefeldt et al., 2016) and are projected to increase with climate change. Recent observations suggest that even very cold permafrost with near surface excess ice is highly vulnerable to rapid thermokarst development and

degradation (Farquharson et al., 2019). If this additional process were included within the CMIP6 models it is likely that the sensitivity of permafrost to increase in global mean temperature may be greater than currently projected (Turetsky et al., 2019).

The models project a loss of permafrost under future climate change of between 1.8 and 3.0 x $10^6$ km$^2$/$^oC$. A more impact relevant statistic is the increase in thawed volume (4.3 to 5.5 x $10^3$ km$^3$ /$^oC$) or 20–30 %/$^oC$. This can be used to quantify the

soil carbon made vulnerable to decomposition under climate change. Assuming there is approximately 827 Gt C in the top 2 m





of the permafrost soils Hugelius et al. (2014) and the median maximum summer thaw depth over the entire permafrost region is 1 m, the carbon in the top 2 m of permafrost is 415 Gt C. This means that 80–120 GtC is made vulnerable to decomposition per $^oC$. This is a committed carbon loss and is slightly less than that suggested by Burke et al. (2018): 225–345 Gt C at $2^oC$ $GMT$ change.


*Data availability.* CMIP5 multi-model ensemble data was down loaded from https://esgf-node.llnl.gov/projects/cmip5/ and CMIP6 multi-model ensemble data was down loaded from https://esgf-node.llnl.gov/projects/cmip6/.

*Author contributions.* EJB and GK designed the analyses and EJB carried it out. ZY processed the relevant site observations. EJB prepared the manuscript with contributions from all co-authors.

*Competing interests.* No competing interests

*Acknowledgements.* We acknowledge the World Climate Research Programme, which, through its Working Group on Coupled Modelling, coordinated and promoted CMIP5 and CMIP6. We thank the climate modeling groups (listed in Table 1 and Table S2.1) for producing and making available their model output, the Earth System Grid Federation (ESGF) for archiving the data and providing access, and the multiple funding agencies who support CMIP5, CMIP6 and ESGF. For CMIP5 the U.S. Department of Energy's Program for Climate Model

Diagnosis and Intercomparison provided coordinating support and led development of software infrastructure in partnership with the Global Organisation for Earth System Science Portals. EJB was funded by the European Commission's Horizon 2020 Framework Programme, under Grant Agreement number 641816, the "Coordinated Research in Earth Systems and Climate: Experiments, kNowledge, Dissemination and Outreach (CRESCENDO)" project (11/2015-10/2020) and the Met Office Hadley Centre Climate Programme funded by BEIS and Defra.





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
