# Peer review of "Evaluating permafrost physics in the CMIP6 models and their sensitivity to climate change"

_The Cryosphere, 2019_

## Referee Comment (RC1) · Anonymous Referee #1 · 6 Mar 2020

Overall the authors present an impressive study, which: 1) compares simulated permafrost dynamics in the CMIP6 models with an extensive collection of observational data sets, 2) assesses differences in model performance between CMIP5 and CMIP6, and 3) uses the CMIP6 models to derive a relationship between future increases to air temperatures and predicted volumetric thaw in the upper 2 m of permafrost-affected soils. The analysis is timely and important, and fits very well within the scope of the journal. The analyses described in the paper are generally rigorous and well-described.

My main suggestion for the paper is that the authors should put more effort into connecting modeling outcomes with real-world consequences. Currently the Discussion section is very short, and ends abruptly, after presenting a calculation related to the mass of frozen carbon that may thaw as air temperatures rise. The authors should describe what this number means—for example, how does 80-100 GtC compare with the size of other reservoirs, and what processes might cause that thawed carbon to leave the soil?

Additionally, the Discussion could do a better job of describing the limitations of using the present ensemble of CMIP models to predict permafrost and carbon dynamics. Currently, the second paragraph cites a lack of simulated thermokarst processes as a limitation to earth system models. I think it would be worth describing in a bit more detail how thermokarst might change climate projections, beyond accelerating permafrost degradation. For example, Liljedahl et al. (2016) describe how thermokarst in terrain with ice wedges often results in greater runoff and drier soils, which in turn may impact carbon fluxes out of the active layer. I would like to see a bit more discussion of how processes like this may contribute to climate model uncertainty.

Even outside of thermokarst-affected terrain, there is evidence that melting of excess ground ice affects permafrost thaw in ways that are difficult to capture in earth system models. For example, Shiklomanov et al. (2013) describe how thawing in the upper permafrost may cause uniform subsidence at the landscape scale, which isn't reflected in traditional ALT measurements such as those provided by CALM. This topic is worth mentioning in the Discussion as well, as it adds uncertainty to assessments of model performance against observational data.

Otherwise, most of my comments are related to presentation. The structure of the paper is logical, but at times the text is somewhat wordy and hard to follow. I make some suggestions for more specific changes in my detailed comments, listed by line number, below.

19-21)   Please provide a bit more detail about how warming permafrost will impact each of these processes or systems. For example, it's more descriptive to say that fires in permafrost-affected areas will increase, rather than saying fires will be impacted.

33)   Change to "within some of the models".

43)   Delete the word "being"

35)       This sentence sounds premature for the introduction. I'd change it to "We evaluate whether the sensitivity of permafrost to climate change…"

29)       Delete the phrase "In fact"

53-54)    The sentence beginning "Under increased global mean temperature…" is a bit hard to follow. Please rephrase.

56-57)    You are referring to thermokarst in this sentence, but please be more specific. What are examples of landscapes changing in a hard-to-predict manner? This would be a good place to list a few types of thermokarst, and state that they are all caused by melting of excess ground ice.

63-64)    I'm not sure that the sentence beginning with "The assessed soil diagnostics…" adds anything. Please reword it or remove it.

74-75)    What does "high end of the range of future pathways" mean? Does this just refer to radiative forcing, or does it mean something more in the context of a Shared Socioeconomic Pathway?

105)      Whenever you refer to mean annual ground temperature, please specify at what depth. Is this always at the top of the permafrost?

115-117)  The first time I read this sentence it was unclear that you applied to relationship from Chadburn et al. both to observations and individually to each model. Please clarify this.

117-118)  I wasn't sure what this sentence means. I think you are saying PFbenchmark is useful because it decouples the performance of the land surface module from biases introduced by the climate module. Please specify this if it is the case.

138)      Please make sure MAGST is defined the first time you use the acronym.

156)      I think you mean 0.25 m instead of cm.

164)      MAGST is defined here, but it should have been earlier, in line 138.

175)      Did Slater and Lawrence provide a probability of permafrost being present, if mean soil temperature at the deepest level was below 0?

218-220)  Please provide more details about this method. How was the site-specific relationship between D and MAGT derived, and how was it extrapolated across each grid cell with permafrost?

223)      Change "is" to "are."

233-252)  This paragraph is quite long and hard to follow, especially with all the acronyms in it. I suggest breaking it into two or three paragraphs, and making sure each starts with a clear topic sentence.

279-280)  This sentence was confusing. I think you mean that you binned MAAT at 0.5° resolution and calculated the median offset for each bin.

284)      Change "to" to "too."

285)      Please rewrite the sentence beginning "Figure S1.3 shows the variation…"

292)      Delete "to be able" from the phrase "to be able to accurately represent…"

294-296)  The sentence beginning with "The offsets are a function of…" is very unclear. Please rewrite it or break it into two or three shorter sentences.

328)      Change the comma to a semicolon.

409)      At what depth?

458-459)  I recommend listing some of the northern high-latitude processes explored in these papers. Also, consider adding recent work by Langer et al. (2016), Aas

et al (2019), and Nitzbon et al (2019) on representing thermokarst lakes and ice wedges in land surface models.

Figure 3)  If the orange lines come from the same observational data set, why are they different between the two panels?
Figure 5)  Why do you plot the simulated summer and winter offsets separately, but only plot the surface offset for observations?
Figure 6)  Why is this figure missing some of the subplots present in Figure 7?

References:

Aas KS, et al. 2019. Thaw processes in ice-rich permafrost landscapes represented with laterally coupled tiles in a land surface model. *The Cryosphere* **13**, 597-609.

Langer M, et al. 2016. Rapid degradation of permafrost underneath waterbodies in tundra landcapes—Toward a representation of thermokarst in land surface models. *Journal of Geophysical Research: Earth Surface* **121**, 2446-2470.

Liljedahl AK, et al. 2016. Pan-Arctic ice wedge degradation in warming permafrost and its influence on tundra hydrology. *Nature Geoscience* **9**, 312-318.

Nitzbon J, et al. 2019. Pathways of ice wedge degradation in polygonal tundra under different hydrologic conditions. *The Cryosphere* **13**, 1089-1123.

Shiklomanov NI, et al. 2013. Isotropic thaw subsidence in undisturbed permafrost landscapes. *Geophysical Research Letters* **40**: 6356-6361.

---

## Referee Comment (RC2) · David Lawrence (Referee) · 26 Mar 2020

This paper evaluates the simulation of permafrost in the CMIP6 (and CMIP5) models. The overall conclusion is that, perhaps not surprisingly, there isn't a huge difference between CMIP6 and CMIP5 models in terms of their representation of permafrost, with perhaps a small amount of improvement. The paper utilizes existing metrics and introduces several new metrics, including a metric for thawed volume, to assess the models. Though this paper does not generate much by way of new insight into the sources of the modeled biases, it is still a useful exercise to establish the current state of permafrost simulations in coupled climate models. Overall, the paper is well-written and will make a good contribution to the literature.

[Figure]

I mostly have a collection of specific points focused on clarity. However, I list a few more significant points here:

1. The authors made a decision to include results for any model that had uploaded the requisite data for the analysis. I wonder if this is really the best strategy, however. For many models, there are two versions that, as far as I am aware, are very similar. Examples include the BCC, CESM, CNRM, Hadley Centre, GISS, and MIROC models. I would suggest that the authors consider showing results for only one of each of these pairs of models. I didn't study every figure, but in general, what I saw was that slightly different versions of models from the same institution didn't look very different to each other, especially for many of the large-scale metrics assessed here. Doing this would significantly reduce the number of shown models, which would then make the figures and analysis more rapidly accessible to readers. In the limited number of cases where there is a significant difference in model behavior across models from the same center, it could be noted in the text or as a supplemental figure. I realize this would be a decent amount of work to redraw every figure and table, but . . .

2. The LS3MIP community specifically requested land-only simulations so that they could be used in direct comparison to the coupled model experiments. This paper would be stronger and more meaningful if the land-only simulations were included in the analysis. This would allow a separation between the role of climate biases and biases that arise due to representation of land processes. Again, I realize that this would be a significant amount of work, but it is work that someone in the community really needs to do, and this paper would be a perfect place for it.

Minor points:

1. Table 1: NorESM uses CLM5. Does HTESSEL have a version number? All the other land models have version numbers.

2. Line 184: Doesn't the Chadburn et al. (2017) method also supply a probability of permafrost for each grid cell?

[Figure]

3. Line 208: You note that an advantage of the D metric is that it enables taliks to be identified. I agree that this would be an advantage of D, but I think most taliks form at depths deeper than the 2m level, so I believe due to the 2m restriction, most taliks would be missed, if the models are simulating them.

4. Line 220: Maybe make it clear that this is an estimate of 'present-day' Dtot and Ftot.

5. Figure 2: Any idea if the 20-year average biases are 'robust'. That is, to what extent would internal model decadal variability affect these biases? Maybe it is beyond the scope of the paper, but there are lots of large-ensemble papers in the literature that could likely help make at least a qualitative assessment, or a few models have already submitted several ensemble members for the historical period.

6. Figure 2 caption: "and is not available for every ensemble member". I think you mean for each model. Even though you don't really utilize ensembles in this analysis, probably best to keep the terminology correct so as to avoid confusion.

7. Figure 2: Why are you only showing bias against the PFbenchmark and not the CCI-PF data as well?

8. Figure 3: As above, perhaps replace 'ensemble' with 'multi-model'.

9. Would be helpful to show the observational estimates of Sdepth,eff on Figure S1.2.

10. Given the challenges in determining snow depth in observations, I do wonder how accurate the Sdepth,eff dataset really is across the pan-Arctic. Nothing that you can do about this, but maybe should qualify statements here and there across the paper, noting that snow depth can be highly spatially-variable is difficult to measure and/or assimilate due to this strong local heterogeneity due to aspect, snow redistribution, snow-vegetation interactions, etc.

11. Line 306: Sentence starting with "This means" was confusing to me. Restate?

12. In fact, that whole paragraph seems confusing and would benefit from a rewrite.

13. Line 324 and elsewhere: Do you ever define the "MOHC" models? I couldn't find this acronym defined anywhere, but also could have missed it.

14. The degradation of snow insulation from CCSM4/CESM1 to CESM2 is interesting because the fresh snow density parameterization was deliberately changed to more accurately reflect observations and detailed snow models, and which leads to higher snow density and better permafrost and ALT simulation, at least when forced with GSWP3. See discussion in Lawrence et al., 2019 and van Kampenhout et al., 2019, Figure 1. The new parameterization definitely improves snow densities over ice sheets. Would be interesting to see if the CLM5 land-only simulations show similar relationship as in Figure 6 of this paper. Anyway, not really a comment that needs to be addressed, but just makes me wonder what is going on?

15. Lines 325-328: It was unclear to me in the assessment of the MIROC and CESM models what biases you think are canceling out. Reword?

16. Figure 8: Can you clarify what you are only including the obs grid cells where the model simulates permafrost. Seems to me that it makes this figure harder to understand. Perhaps would be better to keep obs same in all plots, but then report the number of sites where the model doesn't simulate permafrost for any given temperature range . . . or something like that.

17. Figure 10: I think NorESM should be hatched.

18. Line 388: I don't understand what you mean by considerable variation here. Variation across models? Also, not clear why the variability in D at warmer temperatures leads to excessive Dtot and underestimated Ftot? Couldn't variability in D go either way in terms of biasing towards Dtot or Ftot?

19. Line 420: Could also consider reporting the projected permafrost loss per degree of warming of permafrost zone temperature, as in Slater and Lawrence (2013). Gets around the Arctic amplification diversity across models problem, but then, debatably,

may not be as policy-relevant.

20. Line 451: '… slightly different models' → 'slightly different set of models'

21. Line 465. It's true that permafrost might thaw more quickly if abrupt that processes were included, but I think the main point of Turetsky et al. is that the carbon consequences of these abrupt thaw processes could be large … even if the actual area affected is actually quite small. Maybe should clarify.

22. I'm not convinced by the back of the envelope calculation at the end of the paper. Implicit in the D diagnostic is the fact that the seasonal thaw length is extended as well as a deepening of the active layer (and possible talik formation, though as noted above, maybe not much talik within 2m of surface). So, not sure that it is appropriate to simply multiply the change in D by the carbon stock to get the C vulnerable to decomposition. It's quite a bit more complicated than that. Koven et al. (2015) attempted to make this calculation, I think. Further, just because carbon is made vulnerable to decomposition, it doesn't mean that that is a committed carbon loss. Some or even a lot of that carbon could stay in the soil for a long time due to the slow decomposition rates in the still cold and moist soils. Perhaps it would be better to just remove this brief analysis.

Refs:

Koven, C.D, E. A. G. Schuur, C. Schädel, T. Bohn, E. J. Burke, G. Chen, X. Chen, P. Ciais, G. Grosse, J. W. Harden, D. J. Hayes, G. Hugelius, E. E. Jafarov, G. Krinner, P. Kuhry, D. M. Lawrence, A. H. MacDougall, S. S. Marchenko, A. D. McGuire, S. M. Natali, D. J. Nicolsky, D. Olefeldt, S. Peng, V. E. Romanovsky, K. M. Schaefer, J. Strauss, C. C. Treat, M. Turetsky, 2015. A simplified, data-constrained approach to estimate the permafrost carbon-climate feedback. Phil. Trans. R. Soc. A, doi.org/10.1098/rsta.2014.0423.

Lawrence, D.M. R.A. Fisher, C.D. Koven, K.W. Oleson, S.C. Swenson, G. Bonan, N. Collier, B. Ghimire, L. van Kampenhout, D. Kennedy, E. Kluzek, P.J. Lawrence, F. Li, H.

Li, D. Lombardozzi, W.J. Riley, W.J. Sacks, M. Shi, M. Vertenstein, W.R. Wieder,, C. Xu, A.A. Ali, A.M. Badger, G. Bisht, M. van den Broeke, M.A. Brunke, S.P. Burns,, J. Buzan, M. Clark, A. Craig, K. Dahlin, B. Drewniak, J.B. Fisher, M. Flanner, A.M. Fox, P. Gentine, F.Hoffman, G. Keppel-Aleks, R., Knox, S. Kumar, J. Lenaerts, L.R. Leung, W.H. Lipscomb, Y. Lu, A., Pandey, J.D. Pelletier, J. Perket,, J.T. Randerson, D.M. Ricciuto, B.M. Sanderson, A. Slater, Z.M. Subin, J. Tang, R.Q. Thomas, M. Val Martin, and X. Zeng, 2019. The Community Land Model version 5: Description of new features, benchmarking, and impact of forcing uncertainty. JAMES, doi.org/10.1029/2018MS001583.

Van Kampenhout, L., J.T.M. Lenaerts, W.H. Lipscomb, W.J. Sacks, D.M. Lawrence, A.G. Slater, and M.R. van den Broeke, 2017. Improving the representation of polar snow and firn in the Community Earth System Model. JAMES, 9, 2583-2600, doi.org/10.1002/2017MS000988.

---

## Author Comment (AC1) · 1 May 2020

Significant points:

1. The models assessed are now limited to one per institution as indeed the results are very similar for different model versions from the same institute. Another advantage of this is that it enables a few of the more recent additions to be included without expanding the figures further.

2. We have briefly looked at the land-hist simulations and produced a few plots. However, we feel that a comprehensive analysis would be beyond the scope of this paper. In particular it will require extensive collaboration with the respective modelling groups to understand their behaviour.

[Figure]

Minor points:

1. We can't find a version number for HTESSEL - https://cera-www.dkrz.de/WDCC/ui/cerasearch/cmip6?input=CMIP6.ScenarioMIP.EC-Earth-Consortium.EC-Earth3. CLM5 added for NorESM.

2. Indeed the Chadburn et al. (2017) method provides a probability of permafrost – this has been included in the references.

3. We have added a sentence to clarify that "the D metric enables taliks to be identified, although this is more relevant considering soils deeper than 2 m"

4. "present day" has been added to line 220.

5. I have had a quick look at UKESM with respect to different ensemble members. There are 16 in the CMIP archives. The supplement shows a few of the results. This is briefly alluded to in the discussion.

6. "ensemble member" has been replaced with "model"

7. Another line has been added to the middle two plots of Figure 2 showing the bias between the two observationally based PF data sets and hence the bias against the models.

8. Again 'ensemble' has been replaced with 'multi-model'

9. Observations of the effective snow depth have been added to Figure S1.2

10. A discussion on the snow depth observations has been added in the 'Large scale snow depth product' section and when discussing the models' ability to represent the snow depth.

11. We have sorted this paragraph

12. We have sorted this paragraph

13. MOHC is removed

14. The figure attached shows the coupled and uncoupled results from CESM(coupled-historical) /CLM (uncoupled-land-hist). Yaxis shows winter offset, i.e. difference between air and soil surface temperature in winter and x-axis is effective snow depth. The data has been binned by winter air temperature – each colour represents the 5 degree C bins around the mean winter air temperature shown in the legend. This is the equivalent of Figure 6. There are some interesting differences between the two model versions.

15. The paragraph discussing MIROC and CESM has been re-written for clarity.

16. Figure 8 has been re-drawn to keep the obs the same and reports the number of sites where the model doesn't simulate permafrost.

17. NorESM is now hatched in Figure 10.

18. This has been re-written for clarity. Figure 9 suggests that the annual mean thawed fraction in the models is typically larger than in the observations. This has now been explicitly stated In the document.

19. This is an interesting comment and one we debated when preparing the Chadburn et al. (2017) paper. Some papers quote per degree of global warming and some per degree of high latitude warming. We would prefer to keep it as per degree of global warming as we think it is more policy relevant.

20. Changed as suggested.

21. This middle paragraph of the discussion has been re-written in response to the other reviewer's comments.

22. Analysis of carbon stocks removed from the end of the paper.

Please also note the supplement to this comment:
https://www.the-cryosphere-discuss.net/tc-2019-309/tc-2019-309-AC1-supplement.pdf

[Figure]

[Figure]

**Fig. 1.** Winter offset vs effective snow depth for CLM/CESM.

**Supplement:**

**UKESM1-0-LL model ensembles (and JULES-ES offline).**

Figure 1 shows the permafrost area defined in two ways. The plot on the left shows a histogram of the permafrost extent for the 16 UKESM ensembles compared defined using the mean annual air temperature (MAAT - 1995-2014) and the relationship between MAAT and probability of permafrost defined by Chadburn et al. (2017) (PFchadburn). The plot on the right shows a histogram the permafrost extent defined using the soil temperature of the deepest soil level (PFsoil).

[Figure]

[Figure]

*Figure 1 - PF extent for the UKESm ensembles*

The spread of values for PFchadburn is between 16.9 and 18.2 million square kilometres (mean 17.5; standard deviation 0.33; standard error 0.08) and for PFsoil 13-14.3 million square kilometres (mean 13.7; standard deviation 0.34; standard error 0.08). Whilst this spread is 6-8 % of the permafrost extent it is still relatively small compared with the differences between models. When the permafrost extent is normalised by the area of land where the air temperature is less than zero the spread of values is small and can be considered a characteristic of the model (Figure 2 below). The mean is 0.66 with a standard deviation of 0.005 (spread 0.65-0.67).

[Figure]

*Figure 2 - PF extent/area tair less than 0 degrees C (See table S1-1 in paper).*

The following relationships also appear to be a characteristic of the model and relatively independent of climate biases. Figure 3 shows the relationship between effective snow depth and winter surface to air offset for the UKESM1-0-LL ensemble members (orange). Figure 4 shows the relationship between MAGT and MAAT. Figure 5 shows the winter, summer and thermal offsets as a function of temperature. These relationships appears to be pretty robust. However, the differences between JULES-ES and UKESM1-0-LL require further analysis.

[Figure]

*Figure 3- Relationship between effective snow depth and winter offset for UKESM1-0-LL and JULES-ES. Equivalent of Figure 6 in paper.*

[Figure]

MAAT ($^{o}$C)

*Figure 4 - Relationship between MAAT and MAGT compared with the CCI-permafrost observations and WFDEI tair. Each UKESM1-0-LL ensemble member is shown on a different figure and the same observations / JULES-ES simulation is shown on each plot. Equivalent of Figure 7 in paper.*

[Figure]

*Figure 5 - The relationship between mean annual air temperature and winter, summer and thermal offsets for UKESM1-0-LL and JULES-ES. (Equivalent of Figure 5 in paper)*

---

## Author Comment (AC2) · 1 May 2020

Many thanks to the reviewers for their detailed set of comments. We have responded below.

Main suggestion 'the authors should put more effort into connecting modelling outcomes with real world consequences' – We initially included in this paper a rough calculation of the potential impact of permafrost thaw on sea level rise and permafrost carbon. We have been asked by the editor (sea level rise) and the other reviewer (permafrost carbon) to remove these. They suggested that our analysis was too rough and required a much more extensive analysis which would be beyond the scope of the paper.

[Figure]

We have reworded the section in the middle of the discussion to expand the discussion on missing processes affecting the hydrology of the permafrost region. This now reads - 'In particular, excess ground ice which exists as ice lenses or wedges in permafrost soils is a key process that is not included in the current generation of CMIP models. Thawing of ice-rich permafrost ground will lead to landscape changes including subsidence, thaw slumps and active layer detachments and large-scale modification of the hydrological cycle \citep{liljedahl2016pan}. These ice-rich thermokarst landscapes are susceptible to abrupt changes and cover about 20 \% of the northern permafrost region \citep{olefeldt2016circumpolar}. Recent observations suggest that even very cold permafrost with near surface excess ice is highly vulnerable to rapid thermokarst development and degradation \citep{farquharson2019climate}. The inclusion of these processes within the CMIP6 models will further perturb the hydrologcial cycle and result in additional permafrost degradation not yet quantified by current generation of climate models.'

19-21 – This section has been expanded to 'This unprecedented change will have consequences for northern hydrological and biogeochemical cycles. For example, it will result in CO$_2$ and CH$_4$ emissions which will have a positive feedback on the global climate \citep{burke2017quantifying}. The ecology of thaw-impacted lakes and streams is also likely to change with microbiological communities adapting to changes in sediment, dissolved organic matter, and nutrient presence \citep{vonk2015reviews}. Conditions are likely to be more conducive to fire with earlier snow melt and drier ground in spring \citep{wotton2017potential}. Furthermore subsidence from thawing permafrost will cause damage to manmade infrastructures \citep{melvin2017climate, hjort2018degrading}; leading to issues with the overall sustainability of northern communities \citep{larsen2014polar}.'

33 – changed

43 – deleted

35 – changed as suggested

27 - deleted

53-54 rephrased to: Gradual thaw will occur as the global temperature increases leading to an increase in both the $ALT$ and the time over which the near surface soil is thawed.

56-57 now reads - Abrupt thaw processes caused by the melting of excess ground ice will also occur with the landscape destabilising and collapsing \citep{turetsky2020carbon}. These thermokarst processes are not currently represented in Earth System Models and are not assessed here.

63-64 – sentence removed

74-75 - SSP5 is a world of rapid and unconstrained growth in economic output and energy use (SSP5), with a radiative forcing of 8.5 W m-2 by 2100. The description of SSP5-8.5 is changed to: 'high end of the range of future pathways with fossil fuel development and updating RCP8.5'

105- MAGT is always defined at the top of the permafrost and is shown in Figure 1. This is added to the text in a couple of places to clarify.

115-117 – This now reads: 'Using the \cite{chadburn2017observation} relationship, we reconstructed a permafrost probability map from the WFDEI estimates of $MAAT$. In addition, we applied the \cite{chadburn2017observation} relationship to the $MAAT$ for each model to estimate a benchmark permafrost distribution specific to each model ($PF_{benchmark}$)'

117 – 118 – This sentence now reads - 'This model-specific $PF_{benchmark}$ can be used to evaluate the land surface module independently of any climate biases in $MAAT$.'

138 – MAGST is now defined

156 – Agreed should be 0.25 m.

164 – agreed and included earlier as well.

175 – Slater and Lawrence did not provide a probability of permafrost being present if the deepest soil temperature was less than 0 – they just assumed permafrost.

218-220 – A figure has been added showing this observed relationship and more detail discussing its derivation has been added to the paragraph.

223 – is changed to are

233-252 – this paragraph is divided up and clarified.

279-280 – This now reads 'This spread was calculated independently for each model by binning the $MAAT$ into 0.5$^{\circ}$C bins and calculating the median value of each offset for each bin'

284 – to changed to too

285 – first 'between models' is removed.

292 - 'to be able' deleted.

294-296 – this now reads 'The offsets are a function of both $MAAT$ (not shown) and $S_{depth,eff}$. A low $MAAT$ and a high $S_{depth,eff}$ gives a bigger offset (see also \cite{wang2016evaluation}).'

328 – this sentence has been reworded in response to reviewer 2.

409 - 'at the lowest model level or $D_{zaa}$' has been added.

458-459 – this now says 'Over the past few years there have been a lot of model developments improving the representation of northern high latitude processes in land surface models (e.g. \cite{chadburn2015impact, porada2016effects, guimberteau2018orchidee, cuntz2018physically, burke2017vertical, hagemann2016soil, lee2014effects}). \cite{chadburn2015impact, porada2016effects} developed a dynamic moss parameterisation which enables the insulation effect of the moss on the permafrost to be simulated. \cite{burke2017vertical} added a vertically resolved soil carbon model to enable the permafrost carbon to be identified and traced through the soil. \cite{lee2014effects} included a representation of excess ice within the soil which will melt in response to climate change. Many of these processes are yet to be included within the climate models.'

Figure 3 – the resolution was not the same for the two orange lines- this has been changed.

Figure 5 – only the observed surface offset was available and was not separated into summer and winter, but the surface offset is dominated by the winter snow. We wanted to include the summer offset to show its relative size compared to the winter offset.

Figure 6 – Not all models reported snow depth in the CMIP archives. We will add that note in the caption.

---

## Author Response (AR1)

**Evaluating permafrost physics in the CMIP6 models and their sensitivity to climate change**

Eleanor Burke[1], Yu Zhang[2], and Gerhard Krinner[3]

[1]Met Office Hadley Centre, FitzRoy Road, Exeter, EX1 3PB, UK.
[2]Canada Centre for Mapping and Earth Observation, Natural Resources Canada, Ottawa, Ontario, Canada.
[3]Institut des Géosciences de l'Environnement, CNRS, Université Grenoble Alpes, Grenoble, France.

**Correspondence:** Eleanor Burke (eleanor.burke@metoffice.gov.uk)

**Abstract.** Permafrost is an  ubiquitous phenomenon in the Arctic. Its future evolution is likely to control changes in northern high latitude hydrology and biogeochemistry. Here we evaluate the permafrost dynamics in the global models participating in the Coupled Model Intercomparison Project (present generation - CMIP6; previous generation - CMIP5) along with the the sensitivity of permafrost to climate change. Whilst the northern high latitude air temperatures are relatively well simulated by the climate models, they do introduce a bias into any subsequent model estimate of permafrost. Therefore evaluation metrics are defined in relation to the air temperature. This paper shows that the climate, snow and permafrost physics of the CMIP6 multi-model ensemble is very similar to that of the CMIP5 multi-model ensemble. The main  differences are that a small number of models have demonstrably better snow insulation in CMIP6 than in CMIP5  and a small number have a deeper soil profile. These changes lead to a small overall improvement in the representation of the permafrost extent.  There is little improvement in the simulation of maximum summer thaw depth  between CMIP5 and CMIP6. We suggest that more models should include a better resolved and deeper soil profile as a first step towards addressing this. We use the annual mean thawed volume of the top 2m of the soil defined from the model soil profiles for the permafrost region to quantify changes in permafrost dynamics. The CMIP6 models  project that the annual mean frozen volume in the top $2\,m$ of the soil could decrease by 10-40 % / $^{o}$C of global mean surface air temperature increase.

**1 Introduction**

Permafrost, defined as ground that remains at or below $0^{\circ}$C for two or more consecutive years, underlies 22 % of the land in the Northern Hemisphere (Obu et al., 2019). Permafrost temperatures increased by $0.29^{\circ}$C $\pm$ $0.12^{\circ}$C between 2007 and 2016 when averaged across polar and high-mountain regions (Pörtner et al., 2019; Biskaborn et al., 2019). This unprecedented change will have consequences for northern hydrological and biogeochemical cycles. For example, it will  result in $CO_2$ and $CH_4$ emissions which will have a positive feedback on the global climate (Burke et al., 2017b). The ecology of thaw-impacted lakes and streams is also likely to change with microbiological communities adapting to changes in sediment, dissolved organic matter, and nutrient presence

25   (Vonk et al., 2015). Conditions are likely to be more conducive to fire with earlier snow melt and drier ground in spring (Wotton et al., 2017). Furthermore subsidence from thawing permafrost will cause damage to man made infrastructures (Melvin et al., 2017; Hjort et al., 2018), leading to issues with the overall sustainability of northern communities (Larsen et al., 2014). The latest generation of the Coupled Model Intercomparison Project (CMIP6 ; Eyring et al. (2016)) provides an opportunity to increase our understanding of these potential impacts un-

30   der future climate change.

[revised manuscript text omitted]

**2 Materials and Methods**

**2.1 CMIP model data**

Historical and future monthly mean data were retrieved for a subset of coupled climate models from the CMIP6 (Eyring et al. (2016); Table 1) and the CMIP5 (Taylor et al. (2012); Table S2.1) model archive. The historical simulations run from 1850/1860  to the end of 2014 (CMIP5 - to end of 2005). The CMIP5 future simulations are based on Representative Concentration Pathways (RCPs; Taylor et al. (2012)) which combine scenarios of land use and emissions to give a range of future outcomes through to 2100. When available, RCP8.5 (high pathway), RCP4.5 (intermediate pathway) and RCP2.6 (peak and decline pathway) are used here. The  CMIP6  projections (O'Neill et al., 2016) are based on  scenarios that combine Shared Socioeconomic Pathways (SSPs)  with updated RCP emission pathways. The most widely used scenarios are SSP5-8.5 ( a fossil fuel intensive development socio-economic pathway, updating RCP8.5), SSP3-7.0 (a "regional rivalry" SSP with unmitigated fossil fuel emissions at the medium to high end of the range), SSP2-4.5 ( a "middle of the road" SSP with an emission scenario updating RCP4.5), and SSP1-2.6 ( a sustainable pathway with low end emissions, updating RCP2.6).

Monthly diagnostics processed are surface air temperature (tas; equivalent to 2-m temperature), snow depth (snd), and vertically resolved soil temperatures (tsl) for latitudes greater than 20°N for the first ensemble member of each model where available (i.e., simulation r1i1p1 or similar). Each model is left at its native grid. In addition, grid cells with exposed ice or glaciers at the start of the historical simulation are masked out and the land fractions in the models are accounted for in any area-based assessment of permafrost.

| Model |  Institute | Land mode |
|---|---|---|
|  ACCESS-ESM1-5 |  CSIRO |  CAB |
|  BCC-CSM2-MR | BCC | BCC_AVIM |
|  CAMS-CSM1-0 |  CAMS |  CoLM |
|  CESM2 |  NCAR | CLM5 |
|  CNRM-ESM2-1 | CNRM-CERFACS | Surfex 8.0c |
| CanESM5 | CCCma | CLASS3.6/CTE |
| E3SM-1-0 | E3SM-Project | ELM v1.0 |
| EC-Earth3 | EC-Earth-Consortium | HTESSEL |
| FGOALS-f3-L5 | CAS | CLM4.0 |
| GFDL-CM4 | NOAA-GFDL | GFDL-LM4.0 |
| GISS-E2-1-G | NASA-GISS | GISS LSM |
|  IPSL-CM6A-LR | IPSL | ORCHIDEE (v2.0, /Carbon/Energy m |
| MIROC6 | MIROC | MATSIRO6. |
|  MPI-ESM1-2-HR | MPI-M, DWD DKRZ | JSBACH3.2 |
| MRI-ESM2-0 | MRI | HAL 1.0 |
| NorESM2-LM | NCC |  CLM5 |
| TaiESM1 | AS-RCEC | CLM4.0 |
| UKESM1-0-LL | MOHC, NERC, NIMS-KMA, NIWA | JULES-ES-1 |

**Table 1.** A summary of the CMIP6 models used in this study including the number of soil layers and the depth of the middle of the bottom soil layer. Also show is  $D_{zaa}$ for the models where the difference in the annual maximum and minimum soil temperatures at the maximum soil depth is less than 0.1°C. CMIP5 models are summarised in Table S2.1.

**2.2 Observational-based data sets**

**2.2.1 Air temperature**

Air temperature observations over land at 2 m were taken from the WATCH Forcing Data methodology applied to ERA-Interim (WFDEI) data set (Weedon et al., 2014).  These were generated by applying monthly bias corrections from Climate Research Unit (CRU) (Mitchell and Jones, 2005)  to the Era-Interim reanalysis data (ECMWF, 2009). Air temperatures are available at 0.5° resolution and were aggregated to monthly and annual means.

**2.2.2 Large scale snow depth product**

This data set consists of a Northern Hemisphere subset of the Canadian Meteorological Centre (CMC) operational global daily snow depth analysis (Brown and Brasnett, 2010). The analysis is performed using real-time, in-situ daily snow depth observations, and optimal interpolation with a first-guess field generated from a simple snow accumulation and melt model driven with temperatures and precipitation from the Canadian forecast model. The analysed snow depths are available at approximately 24 km resolution for the period between 1998 and 2016 and were converted from daily to monthly means. It should be noted that snow depths exhibit high spatial variability and difficult to measure because of land surface heterogeneity. In addition there is little evaluation data available for the Arctic.

**2.2.3 Permafrost extent**

The International Permafrost Association (IPA) map of permafrost presence (Brown et al., 1998) gives a historical permafrost distribution compiled for the period between 1960 and 1990. It separates continuous (90-100%), discontinuous (50-90%), sporadic (10-50%), and isolated ($<$10%) permafrost. This distribution was generated from the original 1:10,000,000 paper map and the version used here was re-gridded to $0.5°$ resolution.

The ESA Climate Change Initiative permafrost (CCI-PF) reanalysis data set (Obu et al., 2019) is a recently developed data set that supplies the mean annual ground temperature (at the top of the permafrost ($MAGT$) and the probability of permafrost for each grid cell. These were derived from an equilibrium model of permafrost at 1 km resolution and provide a snapshot of the 2000–2016 period. The model is driven by remotely-sensed land surface temperatures, down-scaled ERA-Interim climate reanalysis data, tundra wetness classes and a landcover map. These data were within $\sim \pm 2°C$ of in situ borehole measurements. This CCI-PF analysis of permafrost extent is within the range but slightly lower than the estimate of Brown et al. (1998). We use a version of the CCI-PF which has been re-gridded to $0.5°$ resolution.

An alternative way of deriving an observational-based estimate of permafrost presence is to derive the probability of permafrost from the observed mean annual air temperature ($MAAT$). An observational-based relationship was defined by Chadburn et al. (2017) who updated Gruber (2012). The Chadburn et al. (2017) relationship has a 50% chance of the presence of permafrost at -4.3°C. Using the Chadburn et al. (2017) relationship, we reconstructed a  permafrost probability map from the WFDEI estimates of  $MAAT$. In addition we applied the Chadburn et al. (2017) relationship to the $MAAT$ for each model to estimate a benchmark permafrost distribution specific to each model ($PF_{benchmark}$). This model-specific $PF_{benchmark}$ can be used to evaluate the land surface module independently of any climate biases in $MAAT$.

[revised manuscript text omitted]
 between $\widetilde{D}$ and $MAGT$ described in Section 2.2.4 and  shown in Figure 2. The CCI-PF dataset was then used in conjunction with this relationship to estimate $\widetilde{D}$ over the permafrost region. $\widetilde{D}$ is related non-linearly to the  $MAGT$ - the warmer the ground temperature the bigger the annual mean thawed fraction.  Therefore a second order polynomial was fitted to the site-specific relationship between $\widetilde{D}$ and the  $MAGT$–green line in Figure 2. The dashed black lines show the relationship for the 95% confidence intervals. These three curves were then used in conjunction with the  CCI-PF data set to derive the mean and range of $\widetilde{D}$ for each grid cell with permafrost present.  Assuming $z_{max}$ is 2 m and summing over the CCI-PF permafrost area gives a present-day $\widetilde{D}_{tot}$ of 5.6 × 10³ km³ (range 2.1 – 9.5 ×10³ km³ ). $\widetilde{F}_{tot}$ is then 22.2 × 10³ km³ (range 18.4 – 25.8 ×10³ km³).

[Figure]

**Figure 2.** The relationship between the $MAGT$ and $\widetilde{D}$ for observed sites and years where both monthly thawdepths and $MAGT$ are available (Zhang et al., 2018). The green solid line is the best fit and the black dashed lines are the 95% confidence intervals.

**3  Results**

In a global climate model the permafrost dynamics  are affected by both the driving climate and by the  parameterisations used to translate the meteorology into the presence or absence of permafrost namely the land surface module. Here we separate out these two factors and, where possible, identify the relative uncertainties introduced.

250

**3.1 Driving climate**

[Figure]

**Figure 3.** The  climate characteristics of the CMIP6 multi-model ensemble compared with the observations for the period 1995–2014. The air temperature observations are from Weedon et al. (2014); the $PF_{benchmark}$ observations from Chadburn et al. (2017); and the $S_{depth,eff}$ observations are from Brown and Brasnett (2010). The red bars are where the model value is greater than the observations and the blue bars are where the model value is less than the observations.  $S_{depth,eff}$ is for the period 1998-2016 and  has not  been uploaded to the CMIP archive for every model. The green lines represent the difference between the Chadburn et al. (2017) data set and the Obu et al. (2019) CCI-PF data (Table 2).

Figure 3 shows the differences between the observations and the CMIP6 models for relevant climate-related characteristics of the permafrost affected region ( $PF_{aff}$) defined by the CCI-PF data (Table 2). The horizontal grey lines on Figure 3 represent ±15% of the observed value. Absolute values for individual models and the observations are given in Table S1.1. These can also be compared with the CMIP5 multi-model ensemble (Figure S2.1 and Table S2.2).

The  $MAAT$ is, to first order, the driver of the presence or absence of permafrost (Chadburn et al., 2017). The ability of the climate models to correctly simulate the northern high latitudes  $MAAT$ is assessed in the top two panels of Figure 3  (3a and 3b). The CMIP6 models  tend to be biased warm compared with the observations and  the area where the land surface is less than 0°C  is biased low. However, in general the models fall within ±2°C of the observed  $MAAT$ (-6.8°C) and within  ±4.0 × $10^6$ km² of the observed area where  $MAAT$ is less than 0°C (24.4  × $10^6$ km²).  These inter-model differences will be reflected in differences in any estimates of permafrost.

Figure 3c shows how biases in the models' *MAAT* impact the presence of permafrost when permafrost is derived from the *MAAT* using the Chadburn et al. (2017) relationship ($PF_{benchmark}$). $PF_{benchmark}$ ranges between 11.0 and  18.97 × $10^6$ km² (Table S1.2), and the models are fairly equally distributed around the observational-based value of $15.1 \times 10^6$ km² . The green line represents the difference between the Chadburn et al. (2017) observations and the CCI-PF data. Overall the CCI-PF data has less permafrost than both the Chadburn et al. (2017) observations and the majority of models. The differences between models appear smaller when  $PF_{benchmark}$ is normalised by the area where  $MAAT$ is less than 0°C. These values range between 0.58 and 0.68 (Table S1.2)  compared with 0.62  found from the Chadburn et al. (2017) relationship  using WFDEI *MAAT*. The differences between models shown in Figure 3d are caused by differences in the latitudinal dependence of  $MAAT$ for temperatures between 0 and -7.6°C – the threshold temperatures of permafrost presence/absence and continuous permafrost respectively  suggested by Chadburn et al. (2017).

Figure 4 summarises the CMIP6 multi-model ensemble by showing the multi-model mean probability of  permafrost. The left hand figure defines the permafrost using $PF_{benchmark}$. Any region where there is permafrost using this definition is shaded in purple. Superimposed is the contour plot of probability of permafrost from Obu et al. (2019) with the orange lines the limits of 50% permafrost. In general the continuous permafrost area is well represented as 100% permafrost, meaning that all of the models can represent the area of continuous permafrost. However, the permafrost extends further south in a small handful of models, as might be expected from the spread in Figure 3. Figures for individual models are shown in Figure S1.1. The  $PF_{benchmark}$ for each model can be used as the reference data

for evaluating the ability of the land surface component to appropriately estimate permafrost presence.

[Figure]

**Figure 4.** The left hand figure shows the  multi-model probability of permafrost using Chadburn et al. (2017) relationship for each model ($PF_{benchmark}$ $PF_{benchmark}$). The right hand figure shows the  multi-model probability of permafrost where permafrost is defined by the temperature at the  $D_{zaa}$ or the lowest model level for the models with the shallower soil profile ($PF_{ex}$ $PF_{ex}$). These plots are the mean for 1995–2014. The orange lines are the limits for 50% permafrost from the CCI-PF data (Obu et al., 2019).

The CMIP6 multi-model ensemble can be compared with the CMIP5 multi-model ensemble (Table 3). The standard time periods are slightly different for each  multi-model ensemble: the CMIP6 climatologies are for 1995–2014 the the CMIP5 climatologies are for 1986-2005. Therefore the observed values (except for $S_{eff}$ which covers a more limited time period) are slightly different with the  $MAAT$ for CMIP6 about $0.3°C$ warmer than for CMIP5 and the area where the land surface is less than 0°C is 0.4  $\times 10^6$km$^2$ larger for CMIP5. Overall the two  multi-model ensemble means agree with the observations for the metrics derived from air temperature with the majority of the constituent models falling within $\pm 15\%$ of the observed values. Table 3 shows the CMIP6 models are  comparable with the CMIP5 models.

|  | CMIP6 | | | CMIP5 | | |
|---|---|---|---|---|---|---|
|  | observations (1995-2014) | model ens. mean - obs. | Percentage within ±15% | observations (1986-2005) | model ens. mean - obs. | Percentage within ±15% |
| $MAAT$ (°C) | -6.8 | 0.28 | 33 | -7.1 | 0.18 | 46 |
| area $MAAT$<0°C ($\times 10^6$ km$^2$) | 24.4 | -0.58 | 83 | 24.8 | -0.57 | 84 |
| $PF_{benchmark}$ ($\times 10^6$ km$^2$) | 15.1 | 0.15 | 61 | 15.7 | -0.74 | 80 |
| $PF_{benchmark}$/area $MAAT$<0°C | 0.62 | 0.0 | 100 | 0.61 | -0.04 | 94 |
| $S_{depth,eff}$ (m) | 0.25 | -0.08 | 16 | 0.25 | -0.07 | 20 |

**Table 3.** A summary of the CMIP6 climate evaluation metrics compared with both the observations and CMIP5. Where relevant, statistics are given for $PF_{aff}$ defined by CCI-PF (Table 2). The difference between the multi-model ensemble mean and the observations are shown plus the percentage of the multi-model ensemble within ±15% of the observations. It should be noted that the observed $S_{depth,eff}$ is for the period 1998-2016.

[Figure]

**Figure 5.** $S_{depth,eff}$ for 1999-2014 for the multi-model ensemble mean of the CMIP6 models compared with the CMC observations. The right hand plot shows the differences between the multi-model ensemble mean and the observations. All grid cells with $S_{depth,eff}$ less than 0.02 m are masked.

The precipitation will also affect the presence or absence of permafrost - in particular any snowpack will insulate the soil. The land surface scheme translates snowfall to snow lying on the surface and quantifies its insulating capacity. Therefore biases in both the snow amount and snow physics will influence the snow insulation. The snow amount can be represented by the $S_{depth,eff}$. Figure 5 shows the multi-model ensemble median $S_{depth,eff}$ compared with the observations from the CMC snow depth analysis (Brown and Brasnett, 2010) for the time period 1998–2014 and Figure S1.2 shows $S_{depth,eff}$ for the individual models. All grid cells with $S_{depth,eff}$ less than 2 cm are masked. $S_{depth,eff}$ for the Arctic is generally greater than 0.2 m in the multi-model ensemble mean. The observations have some regions at the northern tundra where the effective snow depth is slightly shallower than 0.2 m which are not reflected on the multi-model ensemble mean nor in the individual models. This results in a tendency for the models to slightly over-estimate the snow depth in the tundra. The snow region extends further south in the multi-model ensemble mean than in the observations which reflects some of the variability between models. The large spatial variability in snow over the Arctic will not be well represented by either the models or the observations. Individual CMIP6 models (Figure S1.2 and Figure 3) that notably over-estimate the $S_{depth,eff}$ include BCC-CSM2-MR, EC-Earth3, FGOALS-f3-L, GISS-E2-1-G and IPSL-CM6A-LR. Only 16 % of the models have a mean $S_{depth,eff}$ within ±15% of the observations (Table 3). It should be noted that $S_{depth,eff}$ will be moderated by snow physics and in the case of snow the climate biases cannot be cleanly separated from the land surface biases.

**3.2   Land surface module**

The land surface modules translate the driving climate into the permafrost dynamics. In effect they quantify the offsets shown in Figure 1. Figure 6 shows the spread of these offsets as a function of MAAT for the CMIP6 multi-model ensemble along with an estimate of the observed surface and thermal offsets. This spread was calculated independently for each model by binning the MAAT into 0.5°C bins and calculating the median value of each offset for each bin. The winter offset is by far the largest offset with the largest uncertainty and it is strongly dependent on MAAT. Therefore snow plays a dominant role in the relationship between MAGT and MAAT. The summer and thermal modelled offsets both have a small negative value, cover a smaller range of values, and are only slightly dependent on MAAT. In comparison to the observations and assuming the summer offset is small, the model-simulated winter offsets are possibly slightly too small at the warmer temperatures and slightly too large at the colder temperatures. Figure S1.3 shows the variation is relatively large between the individual models. For example, MPI-ESM1-2-HR has very little difference between the MAAT and the MAGT with all offsets of the order 1°C or less, similarly ACCESS-ESM1-5 has a relatively small winter offset. In contrast a few models have very high winter offsets which reach over 10°C at cold temperatures (UKESM1-0-LL, CAMS-CSM1-0, FGOALS-f3-L and TaiESM1). However, comparing the CMIP5 models with the CMIP6 models (Figures S1.3 and S2.5) suggests that there is a general improvement since CMIP5 when compared with the

observations.

[Figure]

**Figure 6.**  Multi-model ensemble spread of the median winter, median summer and median thermal offsets for the CMIP6  multi-model ensemble. Individual models for the CMIP6 multi-model ensemble are shown as lines and identified in Figure S1.3 and for the CMIP5 multi-model ensemble in Figure S2.5. The observed surface and thermal offsets summarised from the available point data (Zhang et al., 2018) are added in black. Although the observed surface offset is not directly comparable with the separate winter and summer offsets, these are shown for the models to illustrate their relative magnitudes.

**3.2.1 Mean annual ground temperature**

Figure 6 suggests that in order for a land surface module to  accurately represent permafrost it needs to be able to represent the insulating ability of the lying snow. This is assessed in Figure 7 which shows the insulating capacity of the snow in terms of the difference between the winter air temperature and winter  0.2 m soil temperature. The offsets are a function of both  $MAAT$ (not shown) and  $S_{depth,eff}$. A low $MAAT$ and a high $S_{depth,eff}$ gives a bigger offset (see also Wang et al. (2016)). In Figure 7 only grid cells where the winter mean air temperatures are between -25 and -15$°C$ are shown, similarly for the observed sites. This ensures the comparisons are not biased by differences in air temperatures. The available models reflect the general increase in offset with increasing  $S_{depth,eff}$ for the shallow snow and the saturation of this relationship for the deepest  $S_{depth,eff}$ to varying degrees of accuracy.

 A few of the models (FGOALS-f3-L, TaiESM1 and UKESM1-0-LL) have relationships between offset and $S_{depth,eff}$ which are

345    indistinguishable from the observed relationship. The rest of the models have

 an offset at any given $S_{depth,eff}$ which is generally too small, suggesting that these models do not have enough snow insulation. The net impact of the snow offset needs to be interpreted carefully in combination with the  $S_{depth,eff}$

350    in order to evaluate the impact of the snow insulation on permafrost dynamics. Figure 5 shows that  Arctic snow depths are relatively shallow. Therefore, because there is a non-linear relationship between offset and  $S_{depth,eff}$, small differences in  $S_{depth,eff}$ will have a big impact on the insulating ability. The models typically tend to slightly over-estimate $S_{depth,eff}$.

355    ~~better than expected. This will affect the ability of the models to simulate the $MAGT$, particularly in the high arctic where the winter temperatures are very cold. In these regions the permafrost is likely to be continuous. In regions of discontinuous and sporadic permafrost the presence or absence of permafrost will be strongly influenced by the snow insulation. In these warmer areas the relationship between winter offset and $S_{depth,eff}$ is well simulated by nearly all of models.~~ $S_{depth,eff}$ and slightly under-estimate the offset for any given value of $S_{depth,eff}$.

[Figure]

**Figure 7.** Differences between the  air and  soil temperature at 0.2 m for the winter as a function of  $S_{depth,eff}$. Only grid cells/sites where the winter air temperature is between -25 and -15°C are shown. The climatological period of 1995–2014 is shown for the CMIP6 models. The blue points with the error bars are the model data and the dotted black lines and errorbars are the observations derived using the data from Zhang et al. (2018). In addition, only the models where snow depths are available from the CMIP archives are shown.

360    Figure S2.6 shows the equivalent plots for the available CMIP5 models. A similar pattern is observed where the models  are more likely to under-estimate than over-estimate the snow insulation. Although limited availability means it is hard to compare individual models between the CMIP5 and CMIP6 ensemble, specific models can be identified. Specifically CanESM and MIROC show improvements; MRI  and GISS show little change and CESM  and NorESM show some degradation.

365

[Figure]

**Figure 8.**  MAGT as a function of local  MAAT for the CMIP6 models and the climatological period 1995–2014 (in red). The  MAGT observations (in blue) were taken from the CCI-PF data set and the  MAAT from the WFDEI data.

Figure 8 shows the combined impact of the three offsets on the relationship between  MAGT and MAAT compared with an observational-based assessment made using the CCI-PF  MAGT and the WFDEI MAAT. As expected the  MAGT increases with MAAT with the CCI-PF  MAGT approximately 4.5 °C warmer than the WFDEI MAAT. As discussed earlier, this difference is dominated by the winter

370  offset, but the summer and thermal offsets also contribute.  Also shown are the same relationships for the models.  Differences in snow insulation  are reflected here. For example,  despite recreating the observed relationship between winter offset and $S_{depth,eff}$, the BCC-CSM2-MR and UKESM1-0-LL models have a much larger difference between  MAAT and MAGT than the observations

375  at the colder temperature, because there is too much snow on the ground in the high  Arctic. CESM2

has a very similar relationship between  $MAAT$ and $MAGT$ as the observations (Figure 8) despite having a smaller winter offset than is observed for any given value of $S_{depth,eff}$ (Figure 7). However, it has a larger than observed $S_{depth,eff}$ which increases the insulation and ensures a good relationship between $MAAT$ and $MAGT$ in Figure 8. A comparison with the CMIP5 multi-model ensemble (Figure S2.7) shows similar differences. It should be noted that this CCI-PF estimate of  $MAGT$ is a model-derived reanalysis and the observational uncertainties are likely under-estimated in the current analysis.

**3.2.2 Active layer thickness**

[Figure]

**Figure 9.** Active layer thickness ($ALT$) as a function of local  $MAAT$ for the CMIP6 models (in red) and the climatological period 1995–2014. Observations of active layer (in blue) are from the CALM sites and the air temperatures are from the large scale WFDEI data set.  The percentage of the  observed sites which also have permafrost  in the models is shown in each sub-plot.

[revised manuscript text omitted]

    Figure 11 also shows $\widetilde{D}_{tot}$ and $\widetilde{F}_{tot}$ for each CMIP6 model. These metrics are calculated for the present-day permafrost region defined by the model specific  multiplied by 2 m.  $PF_{ex}$. Any differences between models are caused by a combination of differences in the  $MAGT$ and differences in the rela-

445 tionships between $\widetilde{D}$  and $MAGT$. These lead to the considerable  spread in $\widetilde{D}_{tot}$ and $\widetilde{F}_{tot}$ shown in Figure 11. There is a tendency for the models to over-estimate $\widetilde{D}_{tot}$ and under-estimate $\widetilde{F}_{tot}$. This arises partly because of the  higher likelihood of the larger values of $\widetilde{D}$ at the warmer air temperatures, i.e. in the discontinuous and sporadic permafrost (Figure 10). The CMIP6 multi-model ensemble has a similar uncertainty to that of CMIP5 suggesting little improvement in the quantification of $\widetilde{D}_{tot}$ and $\widetilde{F}_{tot}$ between ensembles.

[Figure]

**Figure 11.** (a)  Modelled permafrost extent; b) annual thawed volume ($\widetilde{D}_{tot}$); and (c) and annual frozen volume ($\widetilde{F}_{tot}$) of the top 2 m soil for different CMIP6 models for the climatological period of 1995-2014. Permafrost extents ($PF_{ex}$) derived using the mean temperature at  $D_{zaa}$ are the red with black hatching, and those derived using mean temperature at the bottom of the modelled soil  profile  are in red  without hatching. The empty bars with black outlines are the  $PF_{benchmark}$ derived from the  relationship of Chadburn et al. (2017). The grey shaded area is the range expected from observations.~~All the results are for the CMIP6 models for the climatological period of 1995–2014. The middle figure shows the annual thawed volume ($\widetilde{D}_{tot}$) of the top 2 m for the model-specified permafrost region and the bottom figure shows the annual mean frozen volume of the top 2 m ($\widetilde{F}_{tot}$) for the model-specified permafrost region. These two should add to the permafrost extent shown in (a) multiplied by 2 m.~~

 ### 3.2.4 Evaluation metrics

This section summarises some basic evaluation metrics which can be applied to  quantify the ability of the land surface  modules to represent permafrost dynamics. Relevant climate-related metrics are summarised in Table 3 and shown in Table S1.1 for individual CMIP6 models and Table S2.2 for individual CMIP5 models. Table 4 shows a summary of the land-related evaluation metrics for the observations and the two different  multi-model ensembles along with the
455 percentage of each multi-model ensemble falling within the range with individual models shown in Table S1.2 and Table S2.3. These metrics should be relatively independent of climate and reflect the behaviour of the land surface module. Of particular note is the $ALT$, a key metric when simulating permafrost dynamics, which is much deeper than the observations for both CMIP5 and CMIP6.

460 There is better agreement with the observations for the climate-related metrics (Table 3) than for the land-surface related metrics  with only a very small number of models falling within the range of the observations . In addition, there is no consistency in model performance - no model performs well for every evaluation metric (Table S1.2 and Table S2.3). Overall, the percentage of the models which fall within the observed range is relatively low with the majority falling outside the range of the observations (Table 4). However, there  are some improvements for
465 all the metrics in the percentage of models that fall within the observed range between the CMIP5 multi-model ensemble and the CMIP6 multi-model ensemble.

**3.3 Future projections**

This paper makes future projections of  $PF_{benchmark}$, $PF_{ex}$, $\widetilde{D}_{tot}$ and $\widetilde{F}_{tot}$ as a function of global
470 surface air temperature change ($GSAT$). The results are shown in Figure 12 for all of the available SSP scenarios. $PF_{benchmark}$ depends solely on the projections of air temperature; $PF_{ex}$ depends on the annual mean soil temperature at the lowest model level or $D_{zaa}$; and $\widetilde{D}_{tot}$ and $\widetilde{F}_{tot}$ depend additionally on the thawed component of the soil. We assume the results are scenario independent and calculate the mean of each diagnostics for each model after binning into 0.1°C global  surface air temperature bins. Anomalies are then calculated with respect to the values where the
475 $GSAT$ change is 0.0 °C. Data are excluded any time the simulated permafrost extent falls below 1.0 $\times 10^6$ km$^2$. For most of the models these relationships are approximately linear for temperature changes up to around 3 °C.

The sensitivity of $PF_{benchmark}$ to increasing $GSAT$ shows a loss of between 3.1 and 3.8 $\times 10^6$ km$^2$/°C (25$^{th}$ to 75$^{th}$ percentile; Table 5) in the CMIP6 multi-model ensemble. This derivation of
480 $PF_{benchmark}$ uses the observed relationship between $MAAT$ and the probability of permafrost from Chadburn et al. (2017) and assumes it is for an equilibrium state. Therefore this sensitivity is highly dependent on the  Arctic amplification in the models. The variability in $PF_{benchmark}$ between the different models and different

| | CMIP6 | |
| --- | --- | --- |
| |  Observed (range) | Ensemble mean (25$^{th}$-75$^{th}$ percentile |
| $PF_{ex}$/area $MAAT$<0°C | 0.62 (0.55 to 0.77) | 0.53 (0.4... |
| $\widetilde{D}_{tot}$/area $MAAT$<0°C | 0.22 (0.20 to 0.25) | 0.34 (0.2... |
| Surface offset (°C; -14°C< $MAAT$ <-2°C) | 5.7 (4.2 to 7.1) | 4.9 (3.4 to  |
| Thermal offset (°C; -14°C< $MAAT$ <-2°C) | 0.03 (-0.15 to 0.15) | -0.32 (-0... |
| ALT (m; -12°C< $MAAT$ <-10°C) | 0.5 (0.4 to 0.8) | 1.61 (0.85 to... |
| ALT (m; -6°C< $MAAT$ <-4°C) | 1.2 (0.6 to 2.0) | 2.8 (1.6 to 2... |

**Table 4.** CMIP6 model evaluation metrics summary compared with observations and CMIP5. Individual CMIP6 models are in Table S1.2 and Table S2.3.

 multi-model ensembles is relatively small with no obvious outliers. The sensitivities fall to the lower end of the equilibrium sensitivity proposed by Chadburn et al. (2017) who present a loss of  $4.0^{+1.0}_{-1.1} \times 10^6$ km$^2$/°C.

485

The sensitivity of  $PF_{ex}$ to increasing $GSAT$ is related to both the climate and the land surface module and has a wider range of values  from 1.7 to 2.7 $\times 10^6$ km$^2$/°C for the 25$^{th}$ to 75$^{th}$ percentile  (Table 5). This range is around 12 to 20 % /°C of the present day permafrost and is  comparable to the CMIP5 sensitivities. The loss of permafrost derived from $PF_{ex}$ is less than from $PF_{benchmark}$. One

490 reason for this is that $PF_{ex}$ includes the interactions of snow and soil thermal and hydrological dynamics. In addition $PF_{ex}$ represents a transient response to $GSAT$. Despite the shallow soil profile in the majority of the models, the methodology used to derive the $PF_{ex}$ means there will be some implicit time delay in the heat transfer from the surface. There are a few outliers in Figure 12b which have very different sensitivities of $PF_{ex}$ to increasing $GSAT$. These outliers include the MIROC  model which projects a low sensitivity of  $PF_{ex}$ to

495 $GSAT$ in both the CMIP6 and CMIP5 multi-model ensembles (see also Figure S2.11). Although there is some improvement in the winter offset in MIROC between CMIP5 and CMIP6 (Figure S2.6 and 7), there is still too little snow insulation in the present day model because $S_{depth,eff}$ is relatively shallow. In addition the slope of the relationship between  $MAGT$ and $MAAT$ is greater than 1. These factors mean MIROC has a 'permafrost-prone climate' (Slater and

Lawrence (2013)).

500

The increase in mean thawed volume ($\widetilde{D}_{tot}$) or the decrease in mean frozen volume ($\widetilde{F}_{tot}$) is actually relatively consistent between the different models and ranges from  2.1 to 5.9 $\times 10^3$ km³ / $^o$C (5$^{th}$ to 95$^{th}$ percentile; Table 5). This represents a mean loss of frozen volume of around 10–40% of the permafrost in the top 2m of the soil per degree increase in  GSAT (5$^{th}$ to 95$^{th}$ percentile). Here MIROC

505  is not an outlier - the sensitivity of $\widetilde{D}_{tot}$ and $\widetilde{F}_{tot}$ to  GSAT in MIROC falls within the spread of the other models and the relationship between $\widetilde{D}$ and  MAGT is comparable with the observed relationship. This suggests that the summer thawing processes are well represented in  MIROC despite some biases in the sensitivity of the mean deep soil temperature to temperature change. In contrast, the IPSL-CM6A-LR model has a much lower sensitivity of $\widetilde{D}_{tot}$ and $\widetilde{F}_{tot}$ to GSAT. This is because it  does not represent the latent heat required for thawing and therefore

510  has too deep an active layer. However, it falls within the spread of the sensitivity of the $PF_{ex}$ to GSAT. This is mainly because it has a very deep soil profile and the $PF_{ex}$ is diagnosed at $D_{zaa}$. These examples illustrate how the parameterisation of permafrost physics can affect the projections in different ways.

[Figure]

**Figure 12.** Projections of (a) loss of permafrost extent defined as $PF_{benchmark}$ $PF_{benchmark}$ derived from the $MAAT$ $MAAT$; (b) loss of permafrost extent defined as $PF_{ex}$ $PF_{ex}$ derived from the soil temperatures; (c) increase in annual mean thawed volume; and (d) loss of annual mean frozen volume as a function of global  surface air temperature change for the CMIP6 models. All the available scenarios are superimposed on one figure and the results binned into 0.1°C global  surface air temperature change ($GMT$ $GSAT$) bins. $PF_{ex}$ $PF_{ex}$ is greater than 1  $\times 10^6$ km$^2$ in all cases.

| | CMIP6 | | | | | | |
|---|---|---|---|---|---|---|---|
| Percentile (%) | 5 | 25 | **50** | 75 |  95 | 5 | 25 |
|  $PF_{benchmark}$/ $GSAT$ ($10^6$ km$^2$/$^o$C) |  -4.8 |  -3.8 |  **-3.5** |  -3.1 |  -3.0 | -4.2 |  -3. |
|  $PF_{ex}$/ $GSAT$ ($10^6$ km$^2$/$^o$C) |  -3.4 |  -2.7 |  **-2.2** |  -1.7 | -0.3 | -3.5 | -3.2 |
| $\widetilde{D}_{tot}$/ $GSAT$ ($10^3$ km$^3$/$^o$C) |  2.1 |  3.0 |  **4.7** |  5.3 |  5.9 |  2.7 |  4. |

**Table 5.** Projections of loss of  $PF_{benchmark}$,  $PF_{ex}$ and $\widetilde{D}_{tot}$ as a function of sensitivity to global  surface air temperature change ( $GSAT$).

**4    Discussion and Conclusion**

This paper examines the permafrost dynamics in both the CMIP6 multi-model ensemble and CMIP5 multi-model ensemble using a wide range of metrics. As far as possible, the metrics were defined so as to identify the effect of biases in the climate separately to biases in the land surface module. Overall, the two  multi-model ensembles are very similar in terms of climate, snow and permafrost physics and projected changes under future climate change. This paper does not attempt to document specific improvements to individual models in any detail  — the CMIP5 and CMIP6 ensembles  contains a slightly different set of models. However, it is apparent that the snow insulation is improved in a few of the models which results in overall less variability in the permafrost extent ( $PF_{ex}$) in the CMIP6 ensemble than in the CMIP5 ensemble. In general, the ability of the models to simulate of summer thaw depths is  little improved between ensembles. One reason for this remains limitations caused by shallow and poorly resolved soil profiles.

Over the past few years there  have been a lot of model developments  that improved the representation of northern high latitude processes in land surface models (e.g.   Chadburn et al. (2015); Porada et al. (2016); Guimberteau et al. (2018); Cuntz and Haverd (2018); Burke et al. (2 ). Chadburn et al. (2015); Porada et al. (2016) developed a dynamic moss parameterisation, which enables the insulation effect of the moss on the permafrost to be simulated. Burke et al. (2017a) added a vertically resolved soil carbon model to enable the permafrost carbon to be identified and traced through the soil. Lee et al. (2014) included a representation of excess ice within the soil which will melt in response to climate change. Many of these processes are yet to be included within the  climate models.

In particular, excess ground ice which exists as ice lenses or wedges in permafrost soils is a key process that is not included in the current generation of CMIP models.  Thawing of ice-rich permafrost ground  will lead to landscape changes including subsidence, thaw slumps and active layer detachments and large-scale modification of the hydrological cycle (Liljedahl et al., 2016; Nitzbon et al., 2020) . These ice-rich thermokarst landscapes are susceptible to abrupt changes and cover about 20 % of the northern permafrost region (Olefeldt et al., 2016). Recent observations suggest that even very cold permafrost with near surface excess ice is highly vulnerable to rapid thermokarst development and degradation  (Farquharson et al., 2019). The inclusion of these processes within the CMIP6 models will further perturb the hydrologcial cycle (e.g., Fraser et al., 2018) and result in additional permafrost degradation not yet quantified by current generation of climate models.

The CMIP6 models project a loss of permafrost under future climate change of between  1.7 and 2.7 $\times 10^6$ km$^2$/$^o$C. A more impact relevant statistic is the  decrease in annual mean frozen volume (3.0 to 5.3 $\times 10^3$ km$^3$ /$^o$C) or  around 10–40 %/$^o$C.  The projections presented here can be used to ~~quantify the soil carbon made vulnerable to decomposition under climate change. Assuming there is approximately 827 Gt C in the top 2 m of the permafrost soils Hugelius et al. (2014) and the median maximum summer thaw depth over the entire permafrost region is 1 m, the carbon in the top 2 m of permafrost is 415 Gt C. This means that 80–120 GtC is made vulnerable to decomposition per $^o$C. This is a committed carbon loss and is slightly less than that suggested by Burke et al. (2018): 225–345 Gt C at 2$^o$C GMT change.~~ explore the consequences of permafrost degradation on the large scale hydrological and carbon cycles, for example, additional sea level rise (Zhang et al., 2000) and the additional loss of permafrost carbon (Turetsky et al., 2020; Burke et al., 2017b).

[revised manuscript text omitted]